# Atmospheric tomography using the Nordic Meteor Radar Cluster and Chilean Observation Network De Meteor Radars: network details and 3DVAR retrieval

Gunter Stober[1], Alexander Kozlovsky[2], Alan Liu[3], Zishun Qiao[3], Masaki Tsutsumi[4,5], Chris Hall[6], Satonori Nozawa[7], Mark Lester[8], Evgenia Belova[9], Johan Kero[9], Patrick J. Espy[10,11], Robert E. Hibbins[10,11], and Nicholas Mitchell[12,13]

[1]Institute of Applied Physics & Oeschger Center for Climate Change Research, Microwave Physics, University of Bern, Bern, Switzerland
[2]Sodankylä Geophysical Observatory, University of Oulu, Finland
[3]Center for Space and Atmospheric Research and Department of Physical Sciences, Embry-Riddle Aeronautical University, Daytona Beach, Florida, USA
[4]National Institute of Polar Research, Tachikawa, Japan
[5]The Graduate University for Advanced Studies (SOKENDAI), Tokyo, Japan
[6]Tromsø Geophysical Observatory UiT - The Arctic University of Norway, Tromsø, Norway
[7]Division for Ionospheric and Magnetospheric Research Institute for Space-Earth Environment Research, Nagoya university, Japan
[8]University of Leicester, Leicester, UK
[9]Swedish Institute of Space Physics (IRF), Kiruna, Sweden
[10]Department of Physics, Norwegian University of Science and Technology (NTNU), Trondheim, Norway
[11]Birkeland Centre for Space Science, Bergen, Norway
[12]British Antarctic Survey, UK
[13]University of Bath, Bath, UK

**Correspondence:** gunter.stober@iap.unibe.ch

**Abstract.** Ground-based remote sensing of atmospheric parameters is often limited to single station observations by vertical profiles at a certain geographic location. This is a limiting factor for investigating gravity wave dynamics as the spatial information is often missing e.g., horizontal wavelength, propagation direction, or intrinsic frequency. In this study we present a new retrieval algorithm for multi-static meteor radar networks to obtain tomographic 3D wind fields within a pre-defined domain area. The algorithm is part of the Agile Software for Gravity wAve Regional Dynamics (ASGARD) and called 3DVAR, and based on the optimal estimation technique and Bayesian statistics. The performance of the 3DVAR retrieval is demonstrated using two meteor radar networks, the Nordic Meteor Radar Cluster and the Chilean Observation Network De MeteOr Radars (CONDOR). The optimal estimation implementation provide statistically sound solutions and diagnostics from the averaging kernels and measurement response. We present initial scientific results such as body forces of breaking gravity waves leading to two counter-rotating vortices and horizontal wavelength spectra indicating a transition between the rotational $k^{-3}$ and divergent $k^{-5/3}$ mode at scales of 80-120 km. In addition, we performed a keogram analysis over extended periods to reflect the latitudinal and temporal impact of a minor sudden stratospheric warming in December 2019. Finally, we demonstrate the

applicability of the 3DVAR algorithm to perform large-scale retrievals to derive meteorological wind maps covering a latitude region from Svalbard, north of the European Arctic mainland, to mid-Norway.

## 1 Introduction

The mesosphere/lower thermosphere (MLT) is of crucial interest to understand the vertical coupling between the middle and upper atmosphere.Internal atmospheric waves of various spatial and temporal scales drive the MLT dynamics and, thus, provide a considerable source of energy and momentum that is carried from the area of their origin to the altitude of their dissipation as 20 shown by numerous experimental and theoretical studies (Ern et al., 2011; Geller et al., 2013). These waves show strong seasonal and latitudinal/longitudinal characteristics. Gravity waves (GWs) essentially contribute to the MLT energy budget (Fritts and Alexander, 2003; Plougonven and Zhang, 2014) by driving a residual circulation from the summer to the winter pole that results in hemispheric summer mesopause temperatures that are shifted up to 100 K away from the radiative equilibrium (Lindzen, 1981; Becker, 2012). The primary forcing of the residual circulation at the MLT are small-scale GWs arising from 25 various sources in the troposphere. Among them are mountain waves emitted by jet stream imbalances and shear instabilities, frontal systems and GWs excited from deep convection. Recently, theoretical and observational studies put again more emphasize on the role of non-primary GWs, caused by body forces due to breaking and dissipating primary GWs from the lower atmosphere, launching secondary waves into the MLT (Becker and Vadas, 2018; Chu et al., 2018; Vadas and Becker, 2018; Vadas et al., 2018; Heale et al., 2020). Meteor radar measurements of mean winds and momentum fluxes above the southern 30 Andes and the Antarctic Peninsula provided further confidence that non-primary wave generation due to orography plays an important role in the MLT (de Wit et al., 2017; Stober et al., 2021; Dempsey et al., 2021).

Our observational and experimental knowledge of GWs and their dissipation in the MLT has been acquired by remote sensing from ground-based and space-borne sensors, each of which have their own unique observational filters or sensor foot prints (Preusse et al., 2002; Lange and Jacobi, 2003; Ehard et al., 2015; Trinh et al., 2015). Data interpretation ambiguities make 35 comprehensive understanding of the relevant scales, of GW interactions with other waves and the mean flow and resulting energy transfer and wave breaking and dissipation elusive. Satellite observations provide global coverage, but are limited due to their orbit geometry and the line-of-sight (LOS) of their sensors, which often provide limb soundings (e.g., SABER, MLS) and, thus, have constrained capabilities resolving spatial GW characteristics and their temporal evolution. The GW activity and absolute momentum fluxes are often inferred from vertical profiles of temperature (Ern et al., 2011; Trinh et al., 2018) leaving 40 a rather large uncertainty for the horizontal wavelength and gravity wave propagation direction. At the stratosphere there are some satellites observations such as the Aeronomy of Ice in the Mesosphere (AIM) with the Cloud Imaging and Particle Size (CIPS) instrument and the Atmospheric Infrared Sounder (AIRS) with 2D and 3D imaging capabilities to resolve the spatial scale of gravity waves and to obtain direction momentum fluxes (Randall et al., 2017; Ern et al., 2017).

Ground-based instruments such as Rayleigh lidars and radars obtain high-resolution data of temperature, density, and/or wind profiles at a single geographic location (e.g., Baumgarten et al., 2017; Hauchecorne and Chanin, 1980; Hoffmann et al., 2007; Takahashi et al., 2015; Rüfenacht et al., 2018), which again creates ambiguities in deriving the intrinsic GW properties as often only either temperature or wind observations are available. There are some resonance lidars with day and nigh-time capability to simultaneously observe horizontal winds and temperatures at the MLT (Krueger et al., 2015; Wörl et al., 2019). These measurements permits to determine the intrinsic gravity wave properties through a hodograph analysis, modeling or by using polarization relations (Fritts et al., 2002). Spatial information about the GW activity is often obtained from airglow imagers (e.g., Taylor et al., 1997; Hecht et al., 2014; Pautet et al., 2016, 2019, 2021). From the airglow layer, one can derive images of temperature or intensity fluctuations, but still, wind measurements are required to unambiguously resolve the intrinsic gravity wave parameters (Smith, 2014). There are only a few observatories or research facilities in the world with a unique suite of simultaneous common volume observations of winds, temperatures and airglow to determine intrinsic GW properties. However, these observations led to many collaborative studies (Cai et al., 2014; Hecht et al., 2014; Yuan et al., 2016; Cao et al., 2016; Hecht et al., 2018).

Observations of spatially resolved winds are more challenging. Already Browning and Wexler (1968) proposed a so-called velocity azimuth display (VAD), which was applied for meteorological radar observations in the troposphere, and later further developed to the volume velocity processing (VVP) method presented by Waldteufel and Corbin (1979). Both methods fit a first-order approximation to the observations by describing a mean horizontal divergence, relative vorticity, and stretching and shearing deformation inside the observation volume. However, these gradient terms are often difficult to interpret and, thus, the VAD and VVP were mostly used to improve vertical velocity measurements (Larsen et al., 1991) or as a background subtraction for active phased array radars such as the MU radar and MAARSY (Stober et al., 2013, 2018b). These radars are monostatic and, thus, are not suitable for determining the relative vorticity. Recently, these techniques have experienced a short revival and were applied to multi-static meteor radar observations as a simple method to approximate spatially variable wind fields (Stober and Chau, 2015; Chau et al., 2017). Recently, Spargo et al. (2019) presented the first study using a multi-static meteor radar network at Buckland Park in Australia consisting of a monostatic radar and one passive receiver to derive GW momentum fluxes.

In this study, we present a 3DVAR retrieval algorithm, which overcomes many of the limitations of previous analysis routines that retrieved the winds only in independent 2D-layers and required dense observational statistics or were based on the VVP method (Harding et al., 2015; Chau et al., 2017), which is to simple to actually derive geophysical GW parameters. Stober et al. (2018a) already generalized the inversion to retrieve arbitrary wind fields, but still the solutions were confined to a 2D-layer and horizontal winds, and showed a dependency on the apriori in some parts of the domain area.

The new algorithm was developed from scratch and is based on the optimal estimation approach (Rodgers, 2000) and is dedicated to investigate GW dynamics on regional scales using sparse observations. Therefore, the software is called Agile Software for Gravity wAves Regional Dynamics (ASGARD) and includes also a retrieval for the momentum flux (Stober et al., 2021), temperatures (Stober et al., 2017) and wave decomposition (Baumgarten and Stober, 2019; Stober et al., 2019). Here we present the 3DVAR algorithm, as part of ASGARD, that incorporates variable geographic and Cartesian coordinate

grids, optional apriori data, and the possibility to assimilate other data sets. The new algorithm solves for the 3D wind at all grid cells using non-linear error propagation and arbitrary spatial correlations to minimize the apriori dependence of the solutions. Furthermore. we implemented well-known diagnostics from satellite, ground-based radiometers, and lidars, such as the measurement response and averaging kernels, into the meteor radar retrievals (e.g., Livesey et al., 2006; Schwartz et al., 2008; Stähli et al., 2013; Sica and Haefele, 2015; Hagen et al., 2018, and references therein).

The new algorithm is applied to two meteor radar networks to demonstrate its performance using either only monostatic systems or a combination of monostatic and forward scatter passive receivers. The Nordic Meteor Radar Cluster (Nordic, cf. table 1) consists of five monostatic meteor radars located at Tromsø (TRO) Norway, Alta (ALT) Norway, Kiruna (KIR) Sweden, Sodankylä (SOD) Finland, and Svalbard (SVA) Norway. The Nordic meteor cluster is separated into a high-resolution part using only TRO, ALT, KIR, and SOD and an extended network, which includes Svalbard.

The second meteor radar network is called Chilean Observation Network De MeteOr Radars (CONDOR) and employs a monostatic radar at the Andes Lidar Observatory (ALO), a passive receiver at the Southern Cross Observatory (SCO), and another passive receiver at Las Campanas Observatory (LCO). Based on these data sets, we demonstrate the utility of the technique with a series of case studies of body forces, vortices, and other spatial features that are not observable by other techniques so far. Furthermore, we show a keogram analysis of a minor sudden stratospheric warming (SSW) in December 2019 and the latitudinal resolved tidal response. Finally, we demonstrate the possibility to obtain horizontal wavelength spectra as already outlined in Stober et al. (2018a).

The manuscript is structured as follows. The Nordic meteor radar cluster and CONDOR are described in section 2. Section 3 provides a detailed presentation of the wind retrievals, the WGS84 geometry, forward scatter solvers and the optimal estimation technique, which includes the concept of measurement response, variable geographic and Cartesian grids. The performance of the algorithm is demonstrated in section 4 showing examples of body forces, horizontal wavelength spectra, and keogram analysis. The results of the retrieval are discussed in section 5 and conclusion are drawn in section 6.

## 2 Nordic Meteor Radar Cluster and CONDOR

Meteor radar (MR) observations have become a standard tool to investigate MLT dynamics such as winds, tides, GWs and GW momentum fluxes (e.g., Hocking et al., 2001; Holdsworth et al., 2004; Fritts et al., 2010; Pancheva et al., 2020; Spargo et al., 2019; Stober et al., 2021; Dempsey et al., 2021). These radars are designed for continuous and unattended operation under various climate conditions, making them ideal for remote deployments. However, MRs require a rather large field of view of up to 400 km in diameter to detect a sufficient number of meteors, which occur randomly in space and time, to estimate the instantaneous 3D winds.

All MRs used in this study have very similar designs and operation principles, although they are manufactured by two different companies: ATRAD Ltd (Holdsworth et al., 2004) and Genesis Ltd (Hocking et al., 2001). The systems utilize a single Yagi antenna for transmission and on reception an asymmetric cross, a so-called Jones array (Jacobs and Ralston, 1981; Jones et al., 1998), with an antenna spacing of 1 and 2 and 2.5 $\lambda$. Only the MR at ALT in northern Norway employs a receiver array with 1

**Table 1.** Technical parameters of the Nordic meteor radar Cluster and CONDOR (ALO).

|  | TRO | ALT | SOD | KIR | SVA | ALO |
|---|---|---|---|---|---|---|
| Freq. (MHz) | 30.25 | 31 | 36.9 | 32.50 | 31 | 35.1 |
| Peak Power (kW) | 7.5 | 8 | 7.5/15 | 6 | 8 | 48 |
| PRF (Hz) | 500 | 430 | 2144 | 2144 | 430 | 430 |
| coherent integration | 1 | 1 | 4 | 4 | 1 | 1 |
| pulse code | 4-bit complementary | 4-bit complementary | mono | mono | 4-bit complementary | 4-bit complementary |
| sampling (km) | 1.8 | 1.8 | 2 | 2 | 1.8 | 1.8 |
| location (lat,lon) | 69.59°N, 19.2°E | 70.0°N, 23.3°E | 67.4°N, 26.6 °E | 67.9°N, 21.1°E | 78.2°N, 16.0°E | 30.3°S,70.7°W |

and 1.5 $\lambda$ antenna spacing. Further details about the transmitter power, frequency, and experiment settings are summarized in table 1. The passive receiver stations in Chile are also in a Jones array with the standard spacing of 2 and 2.5 $\lambda$. Besides, all CONDOR stations are further equipped with rubidium disciplined GPS to ensure a high frequency and time coherence between the transmitter and the remote receivers.

Figure 1 shows two map projections of the Nordic Meteor Radar Cluster and CONDOR with color-coded elevation. CONDOR stations are located at the windward side of the Andes, the Nordic Meteor Radar Cluster stations are distributed across northern Scandinavia. Eastward winds at both regions favor the excitation of mountain waves that propagate deeply into the thermosphere. Due to local instabilities and momentum deposition, secondary waves are excited.

## 3 Atmospheric tomography - 3DVAR retrieval

Meteoroids entering the Earth's atmosphere are decelerated and heated by collisions with the ambient atmospheric molecules. Some of them, which have sufficient kinetic energy, are vaporized forming an ambipolar diffusing plasma column, which is called a meteor. MRs measure the line of sight or radial velocity of specular meteor trails that drift with the wind. The position of the meteor in the sky relative to the radar is determined by interferometry and typically given as azimuth and off-zenith angles. MRs have wide fields of view and, thus, large observation volumes making them ideal scientific instruments to infer the spatial characteristics of atmospheric parameters.

### 3.1 Meteor radar winds

Instantaneous winds are often derived by applying so-called all-sky fits (Hocking et al., 2001; Holdsworth et al., 2004) for pre-defined time and altitude intervals. Therefore, the data is sorted into altitude-time bins and a zonal and a meridional wind value

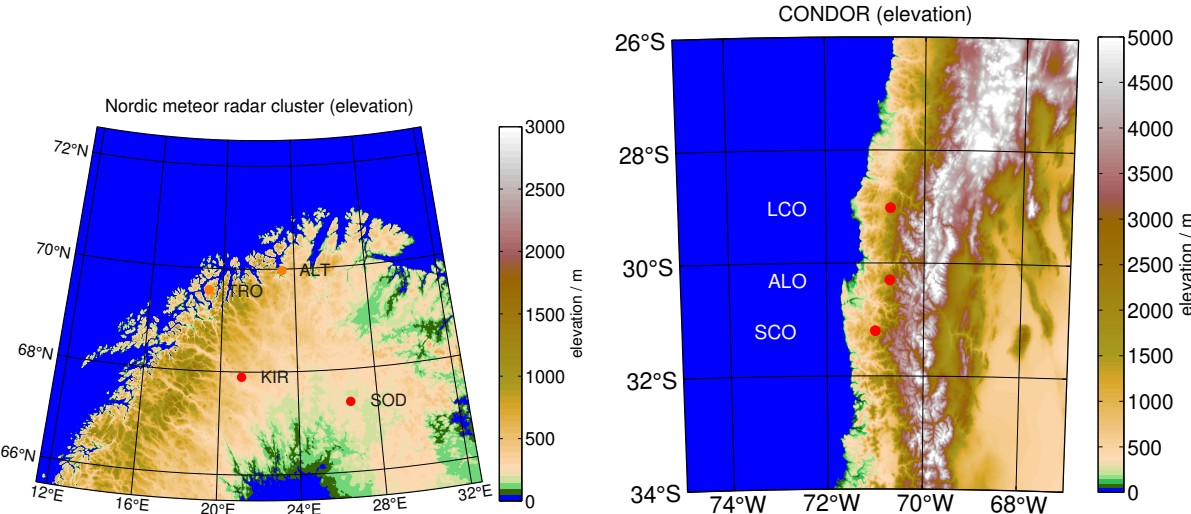

**Figure 1.** Nordic Meteor Radar Cluster and CONDOR locations visualized with color-coded mean elevation. The maps were generated from etopo1 using the m_map package (Amante and Eakins, 2009).

is fitted using least squares by minimizing a function of merit, or cost function, which is given by the radial wind equation;

$$v_r = u\cos(\phi)\sin(\theta) + v\sin(\phi)\sin(\theta) + w\cos(\theta) \quad . \tag{1}$$

Here $v_r$ denotes the observed radial velocity, $u$, $v$ and $w$ are the zonal, meridional and vertical wind components and $\phi$ and $\theta$ denote the azimuth and off-zenith angle at the geodetic coordinates of the meteor (Stober et al., 2018a). An alternative
expression is obtained from the angular Doppler frequency $w_d$ (rad/s), the Bragg vector $\boldsymbol{k}$ and the wind vector $\boldsymbol{u}$;

$$w_d = \boldsymbol{k} \cdot \boldsymbol{u} = u \cdot k_x + v \cdot k_y + w \cdot k_z \quad . \tag{2}$$

Doppler frequency $f_d$ (Hz) and the angular Doppler frequency are related by $w_d = 2\pi f_d$. The Bragg vector is given by $k_B = 2\pi/\lambda_B$ with $\lambda_B = \lambda/2$, where $\lambda$ is the radar wavelength and $\lambda_B$ describes the Bragg wavelength. We will get back to the Bragg notation when discussing the forward scatter geometry for CONDOR. However, the tomographic wind retrieval is making use
of the notation of the radial wind equation only, as it depends on physical and geometric parameters and not on the observational technique or radar specific quantities such as the wavelength or Bragg vector.

Very often MR horizontal winds are obtained under the assumption of $w = 0$ m/s, which is reasonable considering the large observation volume at the MLT of about 400 km in diameter (Hocking et al., 2001; Holdsworth et al., 2004). However, in this study we explicitly consider/solve for the vertical wind and actually estimate the residual bias to assess whether the assumption
of a negligible vertical wind holds. We apply the retrievals as presented in Stober et al. (2018a), which includes a full non-linear error propagation (Gudadze et al., 2019), a spatio-temporal Laplace filter, and a World Geodetic System 84 (WGC84) reference ellipsoid Earth geometry. The spatio-temporal Laplace filter is beneficial to reduce a potential ill-conditioning of the wind fit

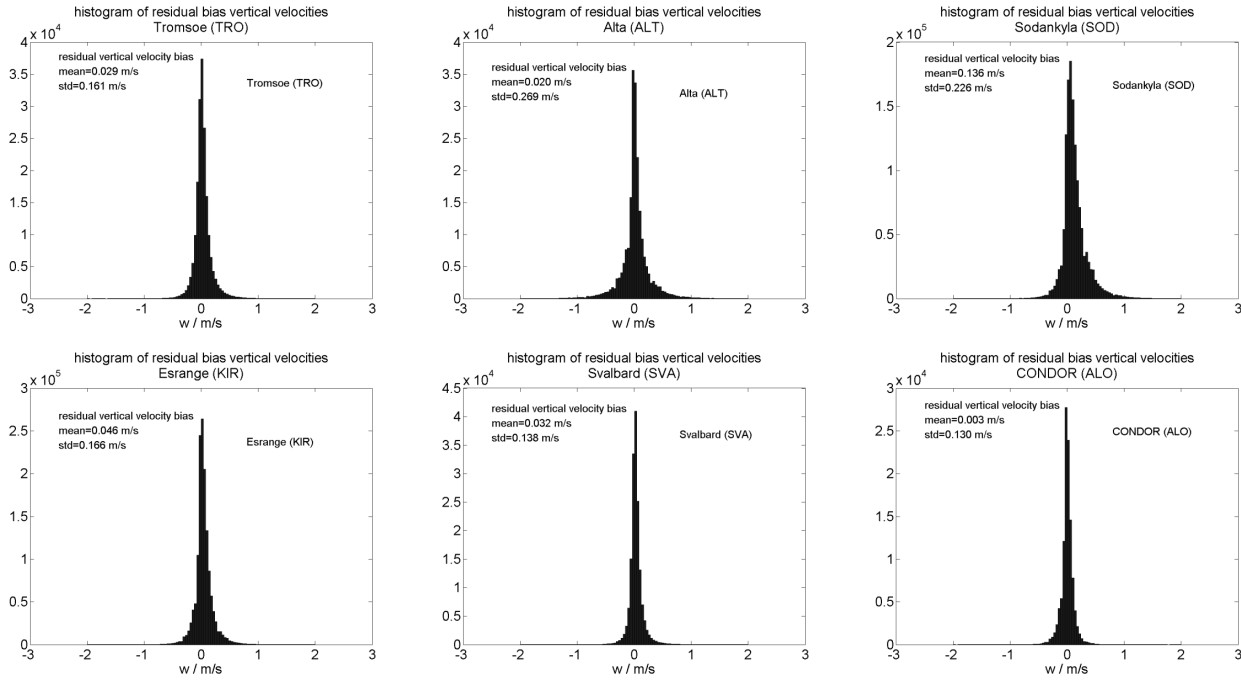

**Figure 2.** Histograms of the residual vertical wind bias for TRO, ALT, SOD, KIR, SVA and ALO MR.

due to the random occurrence of meteors inside the observation volume. Only a few meteors enter the analysis at the upper and lower altitudes of the meteor layer. The spatial distributions there might not allow to derive all wind components with a good measurement response, resulting in large errors and systematic biases. The spatio-temporal Laplace filter uses a derivative in time and altitude to ensure mathematical smoothness and, thus, reduces the ill-conditioning and potential biases. The WGS84 geometry further minimizes altitude and projection errors on the wind components.

The residual bias for the KIR MR is shown in Figure 2. The histogram is estimated from hourly winds and interpreted as mean residual bias due to the lack of a reliable validation possibility. We observed typical biases of a few cm/s and a standard deviation of about 15-25 cm/s. This quality control helps to remove systematic issues in the interferometric solutions caused by mutual antenna coupling by the surrounding radar hardware. Based on these histograms, we accepted only meteor detections within an off-zenith angle of 65° for TRO, SVA, SOD and KIR as well as CONDOR. For the ALT site, we used a more strict criteria of 55° off-zenith angle as we found systematic deviations for meteors at lower elevation. The inclusion of these meteors was leading to a substantial skewness of the vertical velocity histogram towards positive values, which almost disappeared when using the more limited off-zenith angle range.

## 3.2 Implementing the WGS84 reference ellipsoid

Due to the large observational volume covered by the multistatic MR networks of the Nordic Meteor Radar Cluster and CON-DOR, a realistic representation of the Earth's shape is required. Otherwise, large projection errors could result in significant biases. A good approximation of the Earth's shape is given by the World Geodetic System 84 (WGS84) reference ellipsoid (National Imagery and Mapping Agency, 2000). A detailed description of how to implement the WGS84 into the MR wind analysis was outlined in Stober et al. (2018a). The all-sky fits described by Hocking et al. (2001) included a correction for the Earth's curvature of the computed meteor altitudes and wind estimates by using a mean Earth radius and spherical geometry. Here, we briefly summarize how the WGS84 geometry is considered in the wind retrieval. All involved coordinate transformations are given in appendix A (Zhu, 1993; Heikkinnen, 1982). MRs determine the angle of arrival relative to their geodetic geographic position in the East-North-Up (ENU) coordinate frame. Meteors are typically observed at distances between 80 to 220 km for off-zenith angles up to 65°. The Earth's curvature has to be taken into account when estimating the altitude above the Earth's surface (Hocking et al., 2001; Stober et al., 2018a). However, there is also a non-negligible difference between the ENU coordinates at the radar location and the ENU coordinates at the geodetic position of the meteor itself, which has to be considered in the radial wind equation. Therefore, we have performed a coordinate transformation of all line of sight velocities from the ENU coordinate frame of the radar to the ENU coordinate frame of the meteor according to the following steps.

1. transform geodetic radar coordinates into Earth-Centered-Earth-Fixed (ECEF) coordinates using transformation geodetic to ECEF (A1)

2. determine ECEF coordinates of meteor position using transformation ENU to ECEF (A3)

3. transform ECEF meteor position coordinates into the geodetic location of the meteor using transformation ECEF to geodetic (A2)

4. transform observed line of sight velocity into the ENU coordinate frame at the meteor location using transformation ECEF to ENU (A4)

The implementation of the WGS84 Earth geometry turned out to be important for all meteor wind analysis and essentially reduces the projection and altitude biases of the observed meteors. In particular, at mid- and high latitudes these corrections are rather significant and lead to substantial improvements of the wind as well as the vertical profile of ambipolar diffusion and, thus, has an impact on the temperature estimates (Hocking, 1999; Hocking et al., 2001). Furthermore, the vertical wind mean bias and variances are significantly reduced compared to other studies (Egito et al., 2016; Chau et al., 2021; Conte et al., 2021) without including additional damping terms or regularization constraints to the fitted vertical wind as it was the case in previous studies (Stober et al., 2017; Wilhelm et al., 2017). However, our solver preferences a smaller norm, which apparently is associated with small vertical winds. The algorithm was validated against ECMWF-forecast data using the Middle Atmosphere Alomar Radar System (Latteck et al., 2012) and reached a correlation coefficient of 0.984-0.994 and a slope of 0.987-0.998 (Stober, 2020).

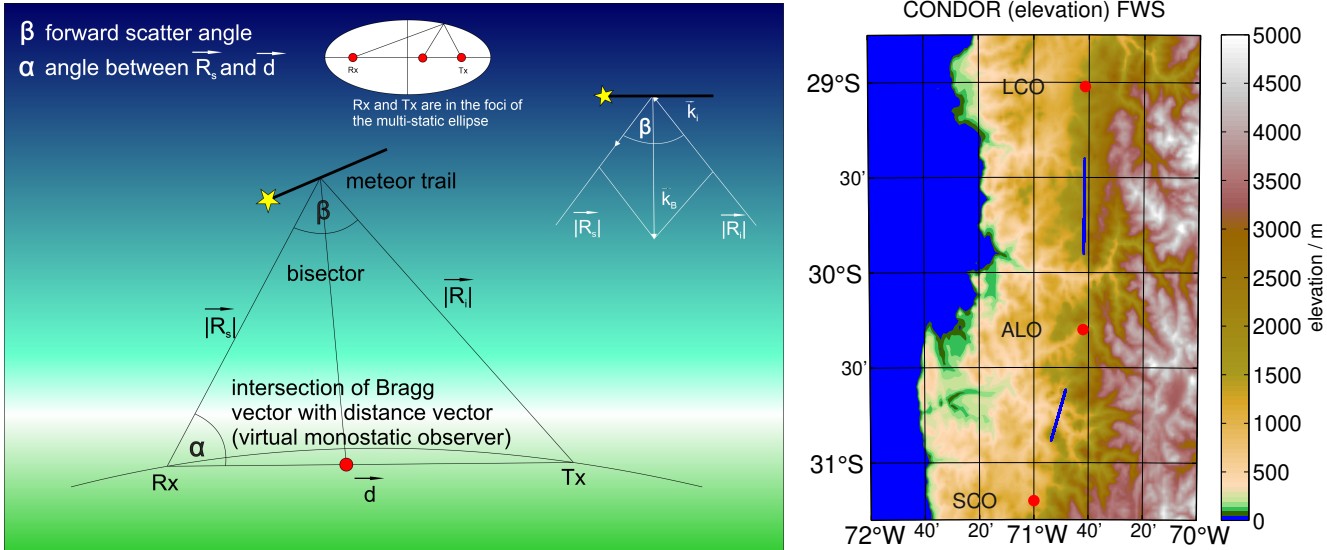

**Figure 3.** Left panel: Schematic representation of the forward scatter geometry and Bragg vector for CONDOR. Right panel: Geographic map of CONDOR with the active and passive radar locations at ALO, SCO and LCO (red dots). The two set of intersection points of the Bragg vectors with the distance vector between Rx and Tx are shown as blue dots, which are only visible as two blue streaks due to the close proximity of the intersection points.

## 3.3 CONDOR-forward scatter geometry

Here, we briefly review the forward scatter position determination for multistatic meteor radar observations such as CONDOR. In ASGARD we updated the algorithm to be more general and applicable to arbitrary meteor radar networks. In Figure 3 we show a schematic representation of the multistatic geometry and a map of CONDOR presenting the locations of the three stations ALO, SCO and LCO (red dots in the right panel) and intersection points of the observed Bragg vectors with the distance vector (small blue dots in the right panel).

CONDOR is located at the Andes and some stations are installed at considerable high altitude. The geodetic positions of the transmitter (Tx) at ALO ($30.251944°$ S, $70.737778°$ W, mean sea level (m.s.l.) altitude 2520.0 m) and the passive receivers (Rx) at SCO ($31.200834°$ S, $71.000000°$ W, m.s.l. altitude 1140.0 m) and LCO ($29.021666°$ S, $70.689720°$ W, m.s.l. altitude 2339.0 m) are known. At each station, the meteor position is determined as the angle of arrival in the ENU frame of reference at the geodetic location of the Rx. The distance vectors $d$ are computed in ECEF coordinates and than transformed into the ENU frame of reference at the Rx sites. The SCO station is about 108.2 km south of ALO and the azimuth ($\phi$) and off-zenith ($\theta$) angles are $\phi = 76.51°$ and $\theta = 89.76°$. The northern CONDOR site at LCO is 136.5 km away from the Tx and the local ENU azimuth and off-zenith angle are $\phi = 268.05°$ and $\theta = 90.54°$. The off-zenith angles are almost, but not exactly, $90°$. The deviations are mostly due to the altitude difference of the Tx compared to the Rx locations and reflects the unique environment for such an installation compared to the multistatic network in Germany (Stober and Chau, 2015; Stober et al., 2018a).

In the next step we solve the multistatic geometry and determine the Bragg vector, which lies within the plane spanned by the meteor, Rx and Tx locations. The angle $\alpha$ shown in Figure 3 denotes the angle between the unit vector from the Rx site to the meteor position in the sky in the ENU frame of reference ($\boldsymbol{R_s}$) and the distance vector to the Tx;

$$\alpha = \arccos \frac{\boldsymbol{R_s} \cdot \boldsymbol{d}}{|\boldsymbol{R_s}||\boldsymbol{d}|} \quad . \tag{3}$$

Furthermore, we measure the total range at the receiver station, which for a monostatic system is 2 times the monostatic range commonly stored in the data stream. However, multistatic links are often further away and thus the signal may travel for a time longer than an interpulse period before reaching the receiver. Therefore, we compute solutions considering a potential range aliasing due to the employed pulse repetition frequency (PRF) of the experiment and convert these into a range $R_0$. The total range then becomes a function of an integer number $n = -1, 0, 1, 2...$ of the measured range $Rge$ and the $R_0$-range;

$$R_n = Rge + n \cdot R_0 \quad . \tag{4}$$

We now solve $n$ times the arbitrary triangle equation for $R_s$;

$$|\boldsymbol{R_s(n)}| = \frac{R_n^2 - |\boldsymbol{d}|^2}{2 \cdot (R_n - |\boldsymbol{d}|\cos(\alpha))} \quad . \tag{5}$$

The result is an ensemble solution for $R_s(n)$, which can be easily converted to potential meteor altitudes by analogy with the algorithm presented above for monostatic cases. However, only one solution is correct and physically meaningful and kept for all further analysis. That is, we only consider solutions in the altitude range between 70 to 120 km and pick the one with the smallest absolute altitude difference to 91 km.

The Bragg vector is derived by transforming the Tx, Rx and meteor positions in ECEF coordinates and by computing the forward scatter angle $\beta$, which is defined as the angle between the vectors $\boldsymbol{R_s}$ and $\boldsymbol{R_i}$. The bisector is given by $\beta/2$ and lies in the same plane spanned by the points Tx, Rx and the meteor and defines the direction of the Bragg vector. The magnitude corresponds to the radial velocity along the Bragg vector. Similar to the range, the standard radar software often provides a monostatic radial velocity measurement $f_d = \frac{2 \cdot v_r}{\lambda}$, which has to be corrected for the forward scatter geometry by;

$$\lambda_B = \frac{\lambda}{2\cos(\beta/2)} \tag{6}$$
$$v_{rad} = f_d \lambda_B$$

here $\lambda$ is the radar wavelength, $v_r$ is the monostatic radial measurement obtained from the Doppler frequency $f_d$ (Hz) and $v_{rad}$ describes the radial velocity along the Bragg vector. By analogy we correct the radial velocity error to account for the forward scatter geometry as well.

Finally, we compute the intersection point between the Bragg vector and the distance vector to perform a sanity check. In Figure 3 (right panel) we show these positions as blue points for three days. All points are found between the Tx and Rx sites and form streaks around the mid-points between the Tx and Rx locations. The longer the distance vector becomes, the longer the streaks.

### 3.4 Optimal estimation 2DVAR/3DVAR wind retrieval

Atmospheric tomography from multistatic meteor radar networks demands sophisticated mathematical approaches because the number of unknowns exceeds the number of observations. There are several strategies to solve such ill-posed and often ill-conditioned problems by assuming a certain smoothness of the solution or other constraints to cure the mathematical rank deficiency of the problem. Optimal estimation is a technique often applied for atmospheric remote sounding and is described in detail in Rodgers (2000). The optimal estimation technique has become a standard tool in radiometry to retrieve atmospheric quantities such as wind, temperature or trace gas concentration (Livesey et al., 2006; Schwartz et al., 2008; Stähli et al., 2013; Hagen et al., 2018; Schranz et al., 2019; Navas-Guzmán et al., 2016).

The optimal estimation technique makes use of Bayes' theorem and presents a general view to all solutions of an inverse problem (Rodgers, 2000). The Bayesian approach relates the posteriori probability density function (PDF) for a given measurement using prior knowledge of the PDF of the state $x$ and observations $y$ before a measurement is made. The forward model maps the state space onto the measurements. These PDFs are often approximated using Gaussian statistics for the measurement error, which appears to be appropriate for many inverse problems.

The 2DVAR/3DVAR wind retrievals are implemented using a posteriori PDF covariance $S_x$ given by;

$$S_x^{-1} = A^T S_y^{-1} A + S_{x_a}^{-1} \quad . \tag{7}$$

Here $S_{x_a}$ is the a priori covariance matrix, which also includes the correlation lengths ($L_2^2$-norm) and has full rank, $S_y$ denotes the measurement covariances and $A$ is the Jacobian of our forward model and is often rank deficient (there are fewer observations than unknowns). Furthermore, we know an a priori state $x_a$ of our forward model, which is updated using the posteriori and measurement covariances, the forward model at the a priori state $x_a$ and the observations $y$ is expressed as;

$$\hat{x} = x_a + S_x A^T S_y^{-1}(y - A x_a) \quad . \tag{8}$$

Considering a typical tomographic domain area with 20 by 20 pixels/grid cells in longitude and latitude, and 20 vertical layers between 70-110 km, results in a large and highly sparse matrix of $S_x^{-1}$, which has a block diagonal shape. Instead of solving such a large matrix by brute force, as a first step, we divide the problem into several smaller blocks. Each block corresponds to a horizontal layer (2DVAR). In a second iteration, we add the vertical coupling for all grid points with observations by adding an additional cost to the inverse problem (3DVAR). This cost is estimated by computing the vertical derivative between the layer above and below or, at the domain boundaries, between the layer itself and the one above or below. Vertical layers without any meteor observations are omitted. We implemented a user defined threshold for the minimum number of meteors in one layer to still be retrieved. This was tested with as low as 6 meteors, but it turned out to be more beneficial to use at least 10 to 15 meteors. In total, we perform 8 iterations mainly to include a non-linear error propagation similar to Gudadze et al. (2019) and to update the 3DVAR vertical cost-function. However, the apriori and apriori covariances are not updated between the iterations.

The forward model in the retrieval is given by the radial wind equation 3.1 in ENU coordinates at the geodetic position of each meteor. The standard error is assumed to be 20 m/s for the horizontal wind components and about 5 m/s for the vertical

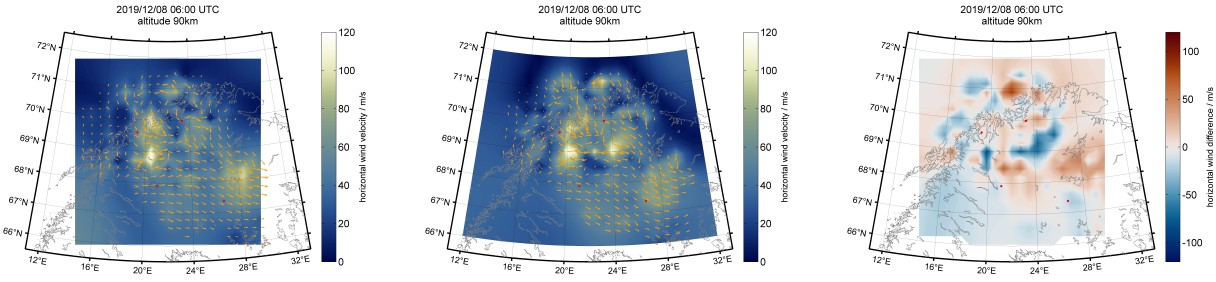

**Figure 4.** Image of 3DVAR retrieved wind fields obtained above the Nordic Meteor Radar Cluster using a Cartesian grid (left panel) and a longitude/latitude geographic grid (central panel). The grey line represents the Nordic coast. The orange arrows denote wind direction and the color bar represents the wind magnitude. The right panel shows a difference plot between both retrievals after remapping the geographic to the Cartesian grid using cubic interpolation.

wind. However, as the choice of the standard error also depends on the measurement statistics and, thus, is a function of the temporal and spatial resolution of the retrievals, we introduced a Lagrange-multiplier to control the smoothness constraint $\alpha^2$. By default the vertical smoothness is 1/4 of the horizontal coupling strength to avoid a too strong instability growth at the upper and lower edge of the meteor layer, when only a few meteors enter the retrieval, and to account for the possibility to use different smoothness constraints and to scale the apriori covariances. Typical values are in the range of $\alpha^2 = 0.02$ to $1$.

We also implemented an option to include longer correlations or polynomial structure functions, which satisfy $L_2-norms$. The software offers to use correlations of $L_2^4, L_2^6, L_2^8$, but the default is always an $L_2^2$-matrix as shown in Stober et al. (2018a). The subscript stands for the norm and the superscript for the correlation length. However, these higher order correlation functions, which are basically spatial derivatives, tend to show an exponential growth at the domain boundaries, which appears to be unrealistic. These functions may be more important when larger domain areas are combined and there is no longer a direct overlap between the observation volumes. Furthermore, the increased spatial correlation is only available in the horizontal domain. The vertical coupling is fixed to the next upper or lower layer, which appears to be most realistic considering the vertical shears induced by atmospheric waves e.g., tidal waves and GWs.

## 3.5 Cartesian and geographic grids

The tomographic retrieval domains can be user defined. Therefore, we implemented two geographic coordinate systems based on a Cartesian and a geographic grid. The Cartesian grid is represented by a geographic longitude and latitude, which defines a xy-coordinate (0,0). All other grid points are measured from this reference in the North-South or East-West direction defining a rectangular grid (Stober et al., 2018a). The geographic grid is given by defining longitude and latitude boundaries and a latitudinal and longitudinal increment. Both grids use as a vertical coordinate the height above the WGS84 reference ellipsoid. Furthermore, we have to define voxels for each grid point. The default voxel is a cylinder with radius $r = \sqrt{2}d$ and height $h$. The variable $d$ denotes the grid resolution for the Cartesian grid. The same voxel shape is applied to the geographic grid, but

the radius is configured to a specific value, which is typically estimated for a certain latitude in the domain to ensure a proper coverage and overlap. Due to the Earth's shape, the effective voxel volume increases with increasing altitude and we keep for each grid cell a fixed longitude and latitude independent of the coordinate grid. The cylindrical shape of the voxel also results in overlapping regions between adjacent grid cells, in particular for the Cartesian grid. However, meteors falling within these overlapping regions are basically detected between two grid cells and, hence, are included in both with the same weight.

In Figure 4 we show two retrieval results using the Cartesian and geographic grid for the same time and altitude and a difference plot after remapping the geographic wind field to the Cartesian grid. The horizontal wind strength is color coded and the orange arrows label the prevailing wind direction for all grid cells, where a meteor was observed. Please note that the wind arrows are always plotted at the grid cell center, which could lead to some shifts of half a grid cell diameter between the two projections. Grid cells that are not driven by observations are just showing the color-coded wind strength and these retrieval solutions are mostly driven by longer correlation lengths and are the result of extrapolations or interpolations. Both retrievals show that the mean wind field is well reproduced and many features are recovered in both grids. Larger dissimilarities occur towards the domain boundaries, where no observations are available. In particular, the body forcing event around the geographic coordinates (69.5°N, 20° E) is found in both images showing an acceleration of the mean flow towards southeast and two counter-rotating vortices indicated by a weakening of the wind strength north and south of the forcing region. Body forces are discussed in more detail in section 4.1. The difference plot shows the variability between both retrievals and provide an estimate on the spatial variance due to the different grids. We obtained a variance of about 20 m/s that is in the order of a typical horizontal wind shear between neighbor grid cells and a median offset of 0.3 m/s between both retrievals considering the whole domain. However, we note that the cubic interpolation between the grids added an additional variability and caused an inflation of the variance that is difficult to assess. Furthermore, we point out that we always plot the wind arrows at the center of a grid cell, which might be not the statistical median position in some part of domain area due to the sparsity of the observations.

### 3.6 Shannon measurement response

Shannon and Weaver (1949) provide the mathematical entropy definition for the information content, which is basically the same as the Gibbs definition of the thermodynamic entropy besides a constant factor k;

$$S(P) = -k \sum_i p_i \ln(p_i) \ . \tag{9}$$

Here $S$ is the entropy and $p_i$ describes the probability of being in state $i$. In thermodynamics $k$ is the Boltzmann constant and in information theory $k = 1$. Considering that we have one state before a measurement and one after, the change in entropy is given by;

$$H = S(p_1) - S(p_2) \ . \tag{10}$$

The information gain during the retrieval is an important measure and has direct practical implications. We basically want to know how the a priori knowledge is improved by the observations. The averaging kernel $R$ is another important quantity as it

contains information about the information correlation and measurement response for each grid cell and retrieved parameter;

$$R = (A^T S_\epsilon^{-1} A + S_a^{-1})^{-1} A^T S_\epsilon^{-1} A \quad . \tag{11}$$

For linear Gaussian PDFs the information entropy change is linked to the averaging kernel by;

$$H = -\frac{1}{2} \ln |\hat{S} S_a^{-1}| = -\frac{1}{2} \ln |I - R| \quad . \tag{12}$$

The measurement response is obtained by integrating all non-negligible contributions from the averaging kernel for a retrieved quantity at a certain altitude and time bin. Examples of such averaging kernels and the corresponding measurement responses have been presented for ground-based radiometer observations (Hagen et al., 2018) and for satellites e.g. the Microwave Limb Sounder (MLS) (Livesey et al., 2006; Schwartz et al., 2008). Although it is desirable to keep the averaging kernels for the 3DVAR wind retrievals, storing a full rank matrix for each retrieved quantity and time-altitude bin is not very practical. Due to the random occurrence of meteors, the measurement response and the corresponding width of the averaging kernel changes for each grid cell and retrieved quantity for a given time-altitude bin.

Furthermore, there is a major dissimilarity between the integral (line of sight) irradiance observations from satellites or ground-based radiometers and the differential meteor observations, which have a well defined location. This has an important implication for the interpretation of the width of the averaging kernel for a given grid cell and retrieved quantity in the 3DVAR retrieval. An increased width of the averaging kernel reflects an interpolation/extrapolation of information from more remote grid cells. From Figure 4 it is obvious that also grid cells at the domain boundary, where no meteors are detected even in the surrounding grid cells, are driven by the observations from grid cells that are further away, implying long correlations and, thus, a still high measurement response when integrating over the width of the averaging kernel. However, we defined an a priori correlation length and, thus, we compute the measurement response only for short correlations. This results in low measurement responses for grid cells with an increased width of the averaging kernel and a high measurement response for grid cells with narrow averaging kernels.

In Figure 5 we show an example of 3DVAR retrieved wind fields and the corresponding measurement response for an observation in March 2020 from CONDOR. The zonal and meridional wind magnitudes are typical for the time of the day and the season. The wind structures are dominated by the diurnal tide. The zonal and meridional winds are presented as color-coded strength. Reddish colors indicate eastward and northward winds while bluish colors show westward and southward winds, respectively. The measurement response is provided as color-coded intensity. A measurement response close to one represents an entirely data driven solution. There may even be measurement responses much larger than one, which point towards even narrower averaging kernels than assumed in the a priori covariance. The orange and yellow lines correspond to measurement responses of 0.7 and 0.4. The whitish colors reflect a high measurement response, whereas bluish colors point towards increased averaging kernels corresponding to a decreased measurement response. However, these points are still driven by the observations rather than the a priori covariance. The measurement response for CONDOR clearly reflects the layout of the meteor radar network with a preference towards north-south. This set up results in a high measurement response for the meridional component above the sites in north-south direction, but a decreased capability to retrieve zonal winds. However, the zonal

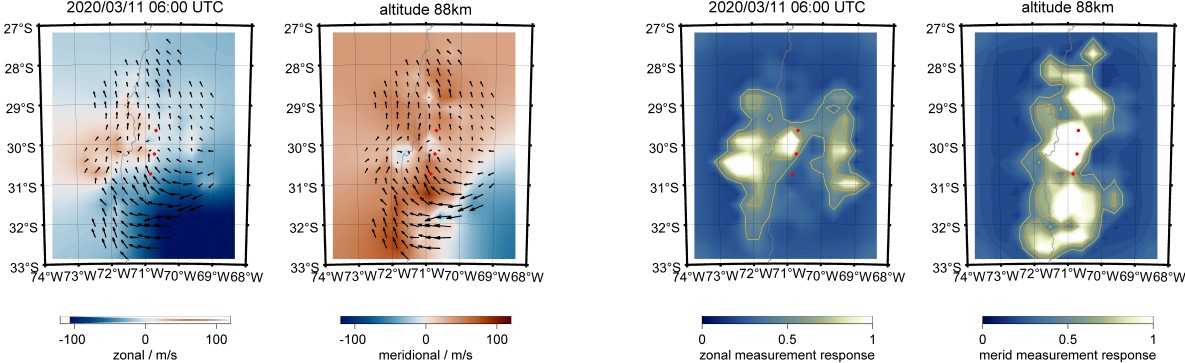

**Figure 5.** The left two panels show the color-coded zonal and meridional wind strength. Bluish colors indicate southward and westward motions, reddish colors highlight northward or eastward winds. Black arrows show the wind direction centered at each grid cell. The right two panels show the color-coded measurement response for the zonal and meridional wind components.

component clearly shows two regions towards the east and west of the radar sites, which indicate a very high measurement response. The measurement response clearly reflects the radial nature of the forward model for the retrieval and the meteor radar network setup. Furthermore, the maps of measurement response indicate the differences between the typical response of monostatic systems compared to passive multi-static receivers. Passive radar links have very distinct sampling responses that are not equivalent to monostatic radars and much less isotropic.

For the Nordic Meteor Radar Cluster an example of the observed 3DVAR wind field and the corresponding measurement response is shown in Figure 6. The map of the zonal and meridional measurement responses reflect again the radial nature of the forward model for the wind retrieval. Meridional winds have high measurement responses north and south of the radar locations and the zonal wind is most reliably retrieved in east-west direction, respectively. Furthermore, the measurement response maps indicate how the different systems complement each other. In particular, TRO, ALT and KIR have a sufficient volume overlap and angular diversity to generate a larger patch with a high measurement response for both wind components (geographic overlap of whitish areas). SOD is located a bit more to the south and east of the cluster and, thus, the typical measurement response is only high for a certain wind component and geographic area.

## 3.7 Data assimilation and a priori state vector

ASGARD offers several possibilities for the a priori state vector. However, considering the optimal estimation technique and the properties of the forward model, which is linear in all coefficients, the final impact of the retrieved parameters appears to be almost independent of the choice of the a priori state vector for all grid cells. This is a major benefit compared to previous

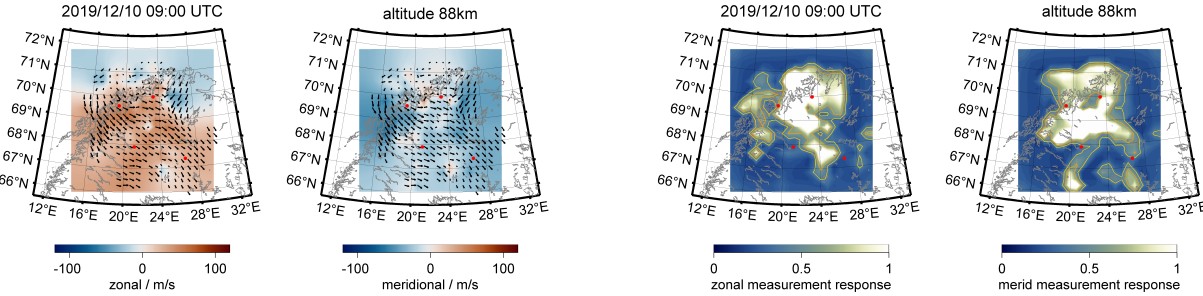

**Figure 6.** The same as Figure 5, but for the Nordic Meteor Radar Cluster.

retrievals presented in Stober et al. (2018a), where the a priori choice of the state vector did impact parameters with a low
measurement response in certain areas of the domain.

There are three options available. The first option is a zero wind state vector for all wind components. The second option is a mean wind vector for the horizontal wind and $w = 0$ m/s. Finally, the third option is a data assimilation wind field, which is estimated from a distance weighted single station wind retrieval or using satellite data such as e.g., TIDI (Wu et al., 2006) assuming again $w = 0$ m/s. A potential satellite data assimilation appeared to have little impact considering the uncertainty in
the geographic location of the observations and the recommended temporal averaging and, thus, was omitted in the current retrievals. However, it is likely worth including over larger or global retrieval domains. All wind fields presented in this paper used the data assimilation approach.

## 4 Science results

### 4.1 Body forces and vortices

Gravity wave dynamics is a vital topic of research. The 3DVAR wind retrievals allow us to study new aspects of spatially and temporally resolved GWs and their interaction with the mean flow. Vadas and Fritts (2001) investigated body forces of breaking GWs and the resulting flow responses with a model. The resulting body forces were characterized by two counter-rotating vortices and a strong acceleration region between them assuming a zero mean background flow.

Recently, there have been several studies on secondary GW generation due to such body forces using a mechanistic model
above the Andes and Antarctic Peninsula (Becker and Vadas, 2018; Vadas and Becker, 2018). These secondary generated GWs resulted in significant changes to the momentum fluxes at the MLT above these regions, which was confirmed by meteor radar observations of the momentum fluxes and wind variances (Dempsey et al., 2021; Stober et al., 2021).

Figure 7 shows a body forcing example of a breaking GW at the MLT above the Andes. Our searching strategy for such events

is based on two data products. First we screen the zonal and meridional winds for enhancements or decreases in the mean flow. In a second step, we investigate the wind residuum to identify two counter-rotating vortices. The wind residuum is estimated by subtracting the mean flow from the images, which is estimated as the median zonal and meridional wind considering the whole domain area. The event in Figure 7 is visible at two altitudes, occurred at approximately 89 km, and had a vertical depth of approximately 3-4 km. The forcing region is indicated by a red circle in panel a) and the yellow arrow points towards the main forcing direction. The body force is deposited in an area of approximately 90-120 km in diameter (3-4 grid cells) and shows a westward acceleration, which leads to a weakening of the strong eastward winds in the forcing region. Panel b) reflects two counter-rotating vortices north and south of the forcing region. The white circular arrows visualize the rotation direction. Panels c) and d) present the same event, but at an altitude of 90 km.

The spatial and vertical dimensions of the vortices and the forcing region is in good agreement with the theoretical work of Vadas and Fritts (2001) and provides confidence in the retrieval results. The vortical structures and forcing region are in areas with a measurement response greater than 0.7 (see Figure 8). We find similar body forcing events for the Nordic Meteor Radar Cluster (see Figure 4).

A major benefit of multi-static observations is the imaging capabilities of large-scale features such as vortices, which remain hidden in monostatic measurements. CONDOR is located at a scientifically interesting region between the low- and mid-latitudes. We very often observe strong regional scale shears in the zonal and meridional wind component associated with vortices. Figure 9 shows a vortex embedded in a large-scale shear flow of a type that is occasionally found above CONDOR. This vortex exhibits a diameter of approximately 300 km. Due to the limited domain size, we are not able to distinguish whether the vortex was the result of a large-scale body force or of a large-scale shear in the zonal component between the equatorial and mid-latitudes.

## 4.2 Large-scale regional dynamics and Keogram analysis

Besides the GW dynamics, the 3DVAR retrieval also captures spatial information of the regional effects of larger scale dynamics such as Sudden Stratospheric Warmings or atmospheric tides and planetary waves. The 3DVAR retrieval results in a significant increase of the available information within the domain area. For a temporal resolution of 1 hour and a vertical resolution of about 2 km between 70-110 km altitude, we obtain about 480 images per day. If we analyze the wind fields at 10 minute resolution, which was occasionally achievable, we generate 2880 images per day. However, for atmospheric tides and planetary waves it is sufficient to look at certain cross sections of the domain area. This can be achieved by keogram analysis, which provides information about the wind dynamics at the temporal evolution for a given longitude and altitude. Here we present Altitude-Time-Wind (ATW) plots, which either show the mean winds over the domain area or at a specific longitude and latitude.

In Figure 10 we show ATWs for the Nordic Meteor Radar Cluster from 20th November 2019 to 15th December 2019, daily mean winds and semidiurnal tidal amplitudes and in Figure 12 the corresponding keograms for the same period at an altitude of 90 km and longitude of 22° E. During this period we observe an early minor stratospheric warming around the beginning of December (bluish colors). Furthermore, there is a strong semidiurnal tidal activity, which enhances after the minor warming

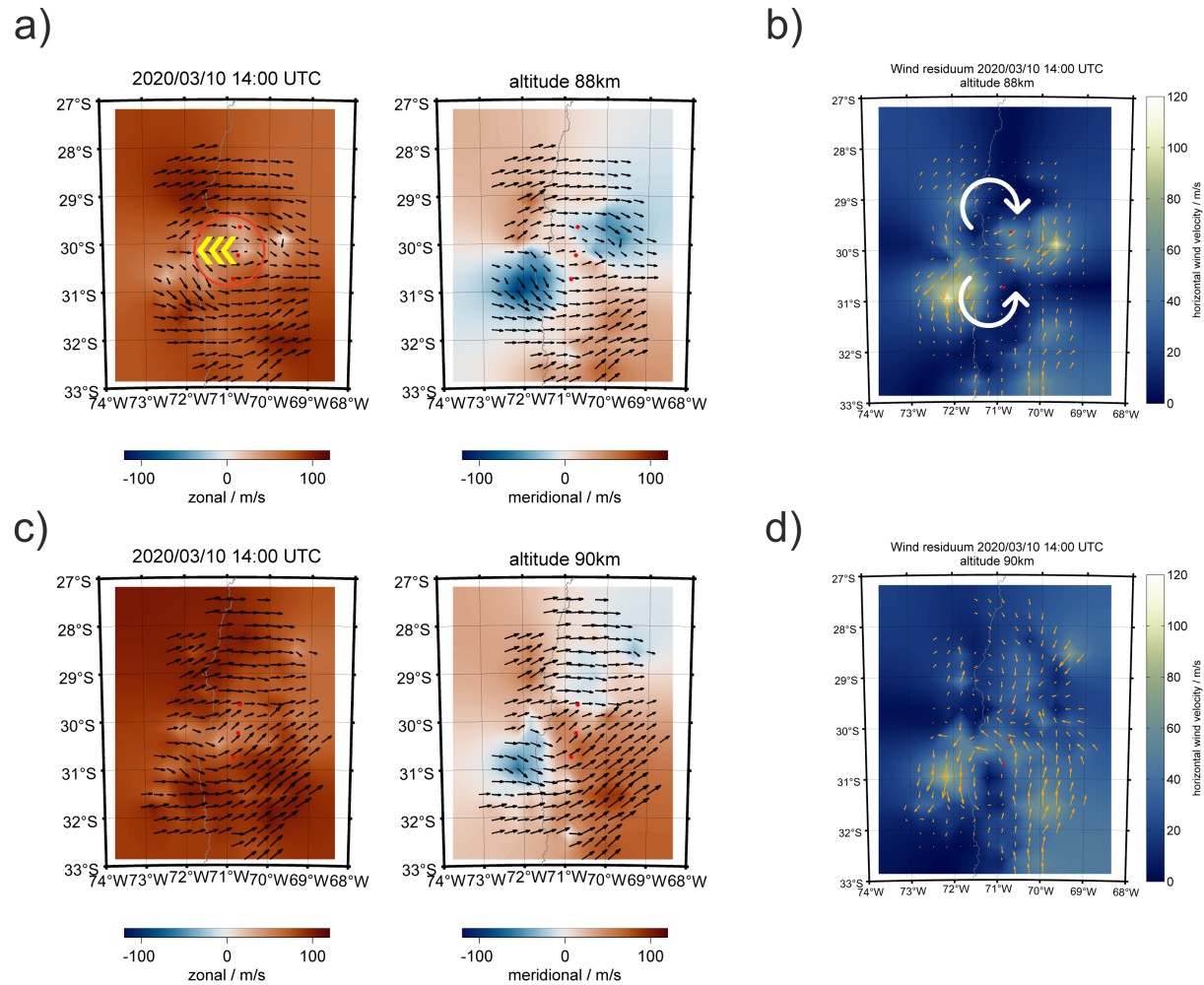

**Figure 7.** Body force of breaking gravity wave above CONDOR. Panels a) and c) show the zonal and meridional components at 88 and 90 km. Panels b) and d) visualize the two counter-rotating vortices after subtraction of the domain mean zonal and meridional wind. The body forcing event is indicated by a red circle, the forcing direction is given by a yellow arrow (panel a) and the two vortices and their rotation direction is presented by white arrows (panel b).

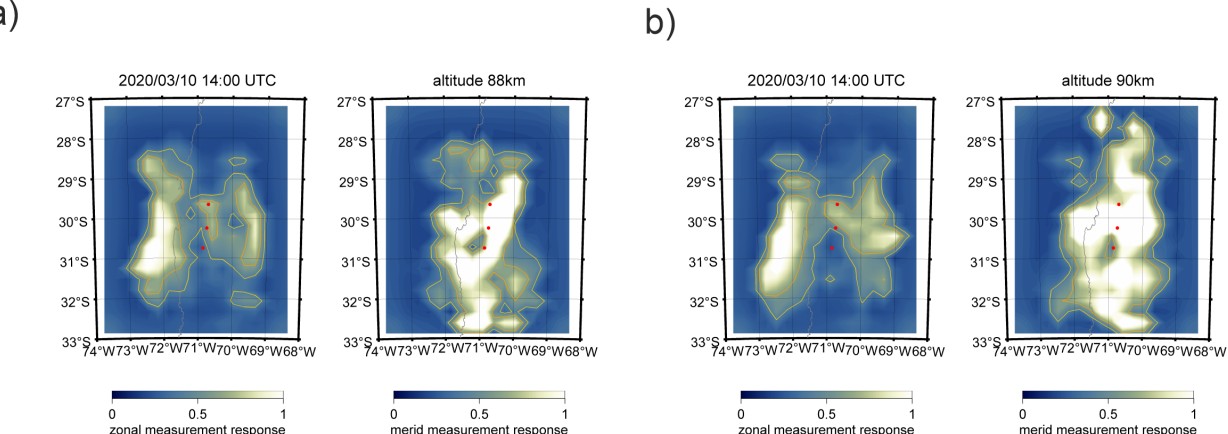

**Figure 8.** The four panels show the measurement response for the zonal and meridional wind component and both altitudes of the body force event above CONDOR.

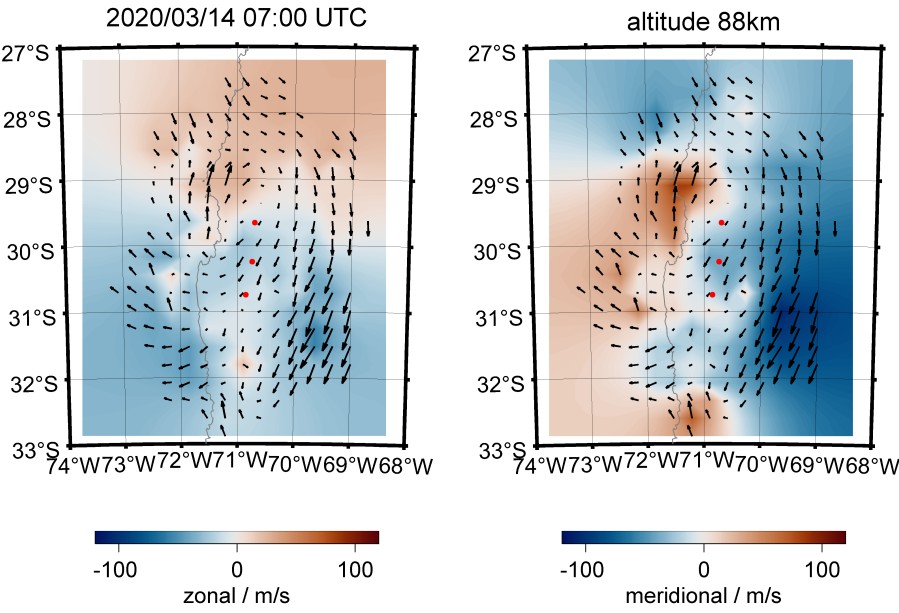

**Figure 9.** The two panels present color-coded 3DVAR retrievals of zonal and meridional winds. The black arrows highlight a large vortex above the Andes observed on 14th of March 2020.

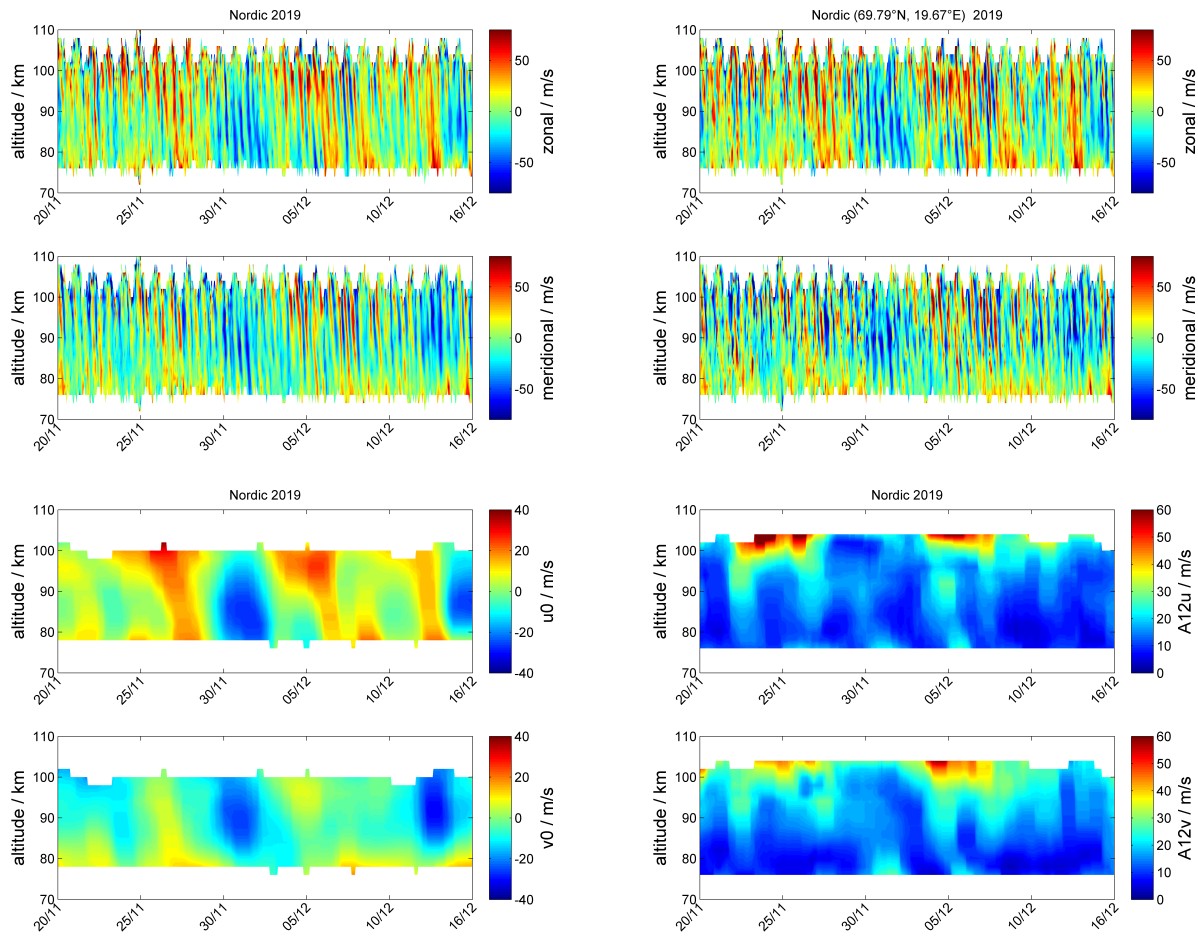

**Figure 10.** The left two panels show Altitude-Time-Wind (ATW) plots for the zonal and meridional color-coded wind strength integrated over the Nordic domain from 20th of November 2019 to 15th of December 2019. The right two panels highlight the zonal and meridional wind strength for the grid cell located around geographic latitude (69.79° N, 19.67°E) for the same period. The lower two rows show the daily mean zonal and meridional wind (left) and the daily semidiurnal tide amplitude for both wind components (right).

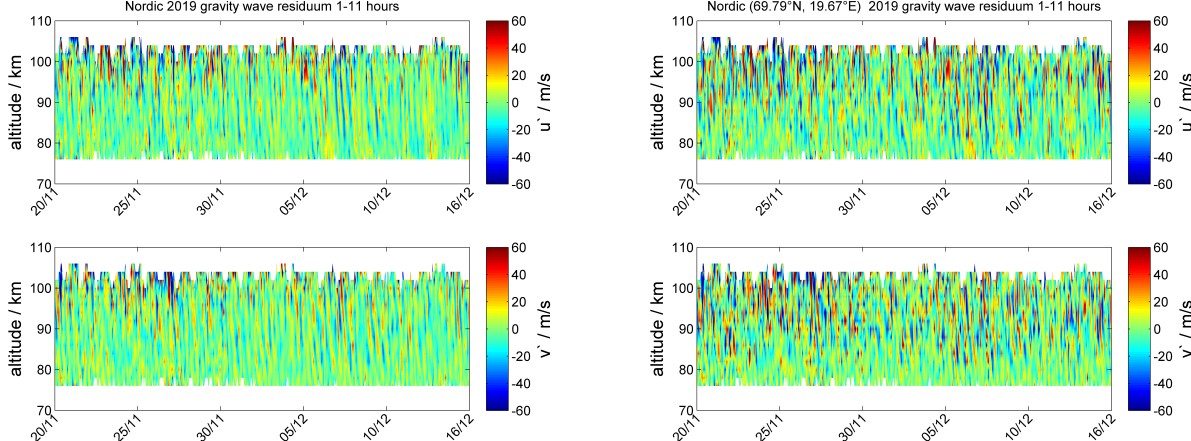

**Figure 11.** The left two panels show the zonal and meridional gravity wave residuum for all waves in the period range between 1 to 11 hours with color-coded amplitude integrated over the Nordic domain from 20th of November 2019 to 15th of December 2019. The right two panels highlight the same but for the grid cell located around geographic latitude (69.79° N, 19.67°E) down to horizontal scales of 60 km.

that occurred at the beginning of December and exhibits an obvious day-to-day variability indicating increased amplitudes during mean eastward winds and inhibited amplitudes for times with more intense westward winds. The left ATW in Figure 10 shows median zonal and meridional winds over the entire domain area, whereas the right panel presents the winds in the column above the grid cell at the geographic location (69.79° N, 19.67° E). Both ATWs reflect the same large-scale dynamical situation with a dominating semidiurnal tide, but the ATW above a single grid cell also highlights the presence of enhanced GW activity, which is removed by averaging the winds over the domain area. Furthermore, Figure 11 presents GW residuals after removing daily mean winds as well as diurnal and semidiurnal tides, which highlights the increased GW activity in the grid cell data compared to the domain average for all waves with periods between 1-11 hours. The filtering was done using the adaptive spectral filter (ASF) approach presented in (Baumgarten and Stober, 2019; Stober et al., 2019).

The keograms in Figure 12 indicate only a weak latitudinal variability of the semidiurnal tidal activity (left panels) and basically exhibits the same dynamical features as the ATWs in Figure 10. The gravity wave activity in the Keograms was examined by subtracting the domain median zonal and meridional wind. At a latitude of about 69-70° N, which is above the Scandinavian mountains, an increased variability is visible (right panels), which might be caused by upward propagating mountain waves. However, further work is required to quantify the spatial variability of the gravity wave activity and to remove additional observational filter caused by the spatial variable measurement statistics. Around 68° S there are more gaps visible in the Keograms compared to the latitudes farther North.

In Figure 13 we show an ATW (whole domain) and a Keogram for the zonal and meridional wind component that was observed with CONDOR during March 2020. The ATW indicates the presence of a strong diurnal tide during this part of the season, which is also seen in the Keogram. the amplitude reaches about 50-60 m/s at altitudes in the range from 90-95 km.

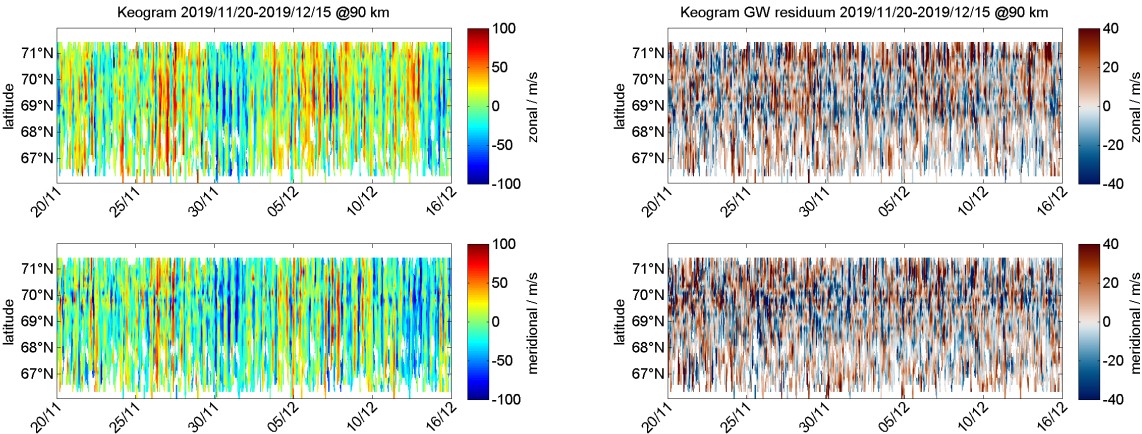

**Figure 12.** The two panels show color-coded latitudinal keograms of zonal and meridional winds for 22° longitude and 90 km altitude from 20th November 2019 to 15th December 2019.

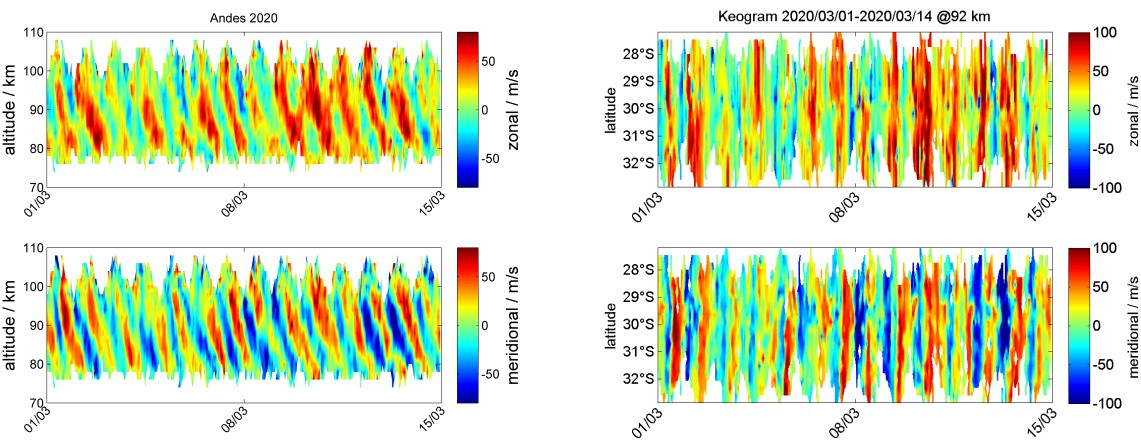

**Figure 13.** The two panels show a ATW measured with CONDOR and the corresponding Keogram for the altitude of 92 km. The data was recorded between 1st to 14th March 2020.

During this 14-day period the diurnal tide exhibits some day-to-day variability similar to what we found at the northern latitudes for the semidiurnal tide. Apparently during the first 8 days of March 2020 the diurnal tide is much weaker compared to the second week. Such a tidal variability seems to be characteristic, However, we did not yet investigate whether a superposition of migrating and non-migrating tides caused this modulation or whether the vertical propagation of the tidal modes was affected in the stratosphere due to gravity waves.

## 4.3 Horizontal wavelength spectra

Another way to investigate the GW activity from the 3DVAR retrievals is to make use of horizontal wavelength spectra, as presented in Stober et al. (2018a). Liu (2019) presents daily zonal wave number zonal wind power spectra derived from Whole Atmosphere Community Circulation Model-Extended at tropospheric and stratospheric altitudes. Nastrom and Gage (1985) estimated kinetic energy spectra from aircraft observations and determined spectral slopes for different wavelength ranges. More recent aircraft observations of horizontal and vertical winds permit to further investigate the spectral shape for each subrange in more detail, but are still limited to the troposphere/lower stratosphere (Schumann, 2019). For horizontal scales between a few km and up to 400-700 km, they derived a spectral slope of $k^{-5/3}$, which is often associated with gravity wave divergent modes, and a slope of $k^{-3}$ for the larger scales. However, there is no observational verification of such kinetic energy spectra at the MLT.

Horizontal wavelength spectra are estimated from the 3DVAR retrievals by computing Lomb-Scargle periodograms (Lomb, 1976; Scargle, 1982) of the horizontal wind along all longitudes for each latitude. Thus, we obtain from each image about 20 horizontal wind spectra at a certain altitude, which results in approximately 480 spectra per day. In Figure 14 we present a typical daily spectrum. The grey points are individual spectral powers from each longitudinal spectra, the black thin line is the daily median spectrum and the cyan and green lines are slopes for a predefined wavelength range that are fitted to the estimated daily median spectrum. The blue and red lines represent spectral slopes corresponding to $k^{-3}$ and $k^{-5/3}$, respectively. We identified a transition scale between the $k^{-3}$ to the $k^{-5/3}$ slope at horizontal wavelengths of about 100-120 km. The slopes at the smaller GW scales were more variable and we found values between $k^{-1.4}$ and $k^{-1.9}$ compared to the larger scales. The smallest resolved scales might indicate steeper slopes than expected due to the spatial correlations given by the coupling strength between adjacent grid cells, which tends to suppress small structures by enforcing a certain smoothness. This would imply that the transition between the vortical-driven scales to the divergent GW scale occurs at much smaller wavelengths than in the troposphere. Nastrom and Gage (1985) estimated the transition scale for the troposphere/lower stratosphere to be around 500-700 km.

## 4.4 Residual vertical velocities 3DVAR

Vertical velocities are also retrieved from the 3DVAR algorithm. However, we are lacking an independent source for validation and, thus, we consider them as an additional quality control rather than a geophysical measurement. The measurement responses for the inferred vertical velocities are rather low, implying that this parameter requires larger correlation lengths compared to the horizontal wind components. We use a priori covariances of 5-7 m/s and the resultant statistical uncertainties are in the range of 2-4 m/s, suggesting at least some improvement of a priori covariances by the observations.

The histograms shown in Figure 15 support at least some statistical interpretation of the variance and median vertical velocity, although the individual values at a certain grid cell may not be useful. These vertical velocity histograms are the result of our forward model and the vertical cost function in the 3DVAR analysis, which forces the vertical velocities to be small by assuming a zero mean value, and depend on the choice of our Lagrange multipliers and the a apriori covariance. The Lagrange

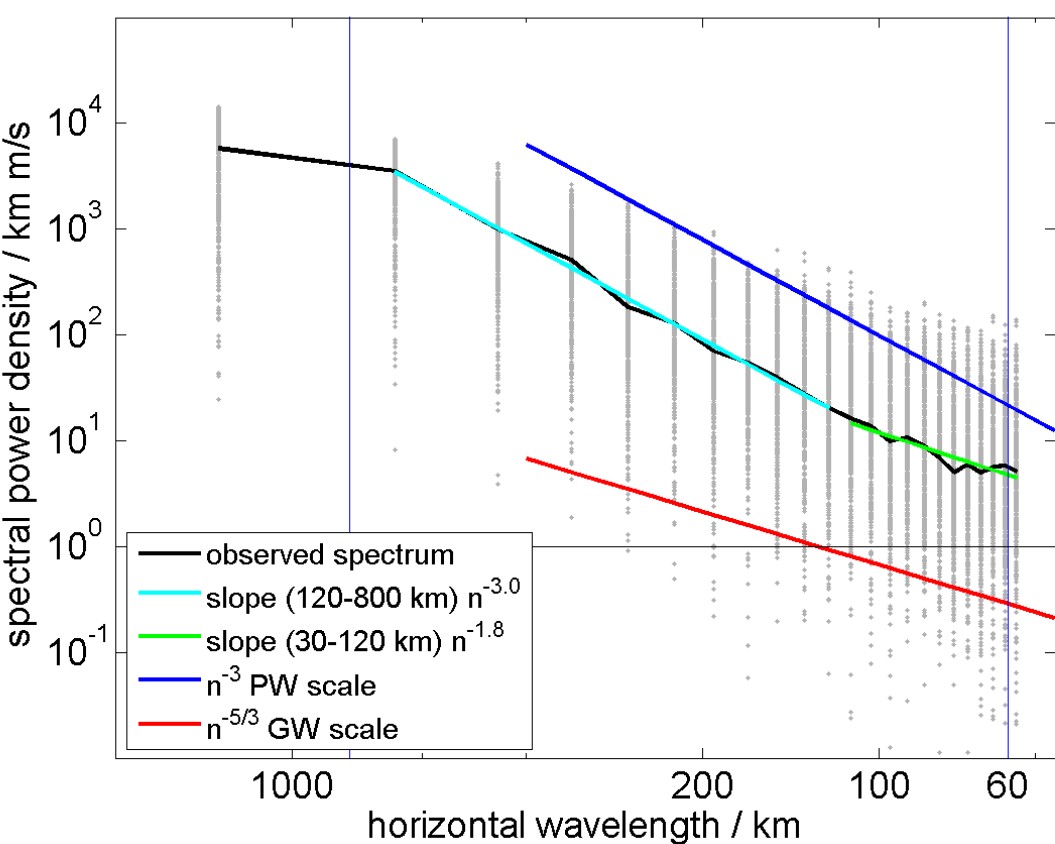

**Figure 14.** Estimated horizontal wavelength spectra for the Nordic Meteor Radar Cluster. The grey points are individual spectral observations, the thin black line is the daily mean spectrum, the cyan and green lines represents fits to the respective wavelengths scale and the blue and red lines are idealized slopes for PW and GW.

multiplier in the current 3DVAR version is set to 1. The results are obtained using a 4 times stronger vertical weight reflecting the smaller covariance in the vertical velocity cost function, which reduces the oscillations or numerical instabilities. Such numerical instabilities can occur in optimal estimation retrievals and are often related to too large apriori covariances, which then result in vertical velocities of up to 100 m/s and/or variances of about 30-50 m/s. Thus, the resulting 3DVAR vertical velocity histograms reflect a statistically sound estimate based on our a priori knowledge rather than an entirely independent

measurement. Please note that the vertical velocities estimated from the standard retrieval in Figure 2 are obtained without these assumptions and there is a high degree of consistency between the distributions concerning the small median value of a few cm/s and the width of the distribution, which scales with the averaging volume. Hence, the variance is about an order of magnitude larger for the 3DVAR retrieval using 30 km diameter grid cells compared to the monostatic winds, which are based on 300 km diameter volumes.

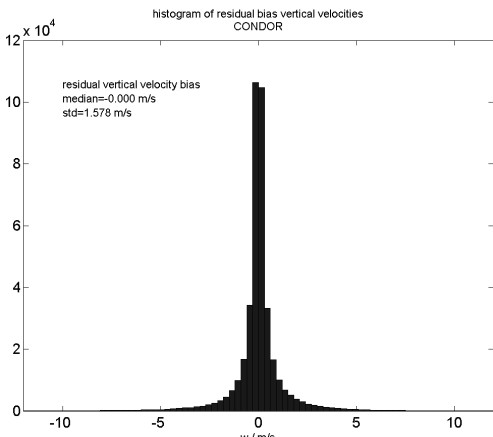
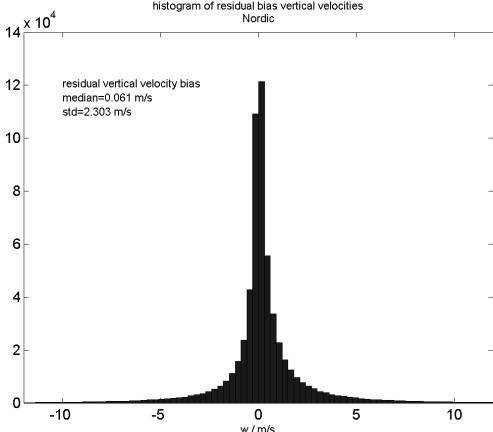

**Figure 15.** Histograms of the residual vertical velocity bias derived from the 3DVAR retrieval for three successive days and integrated over the full 3D domain area.

## 5   Large-scale retrievals – meteorological wind maps

Finally, we demonstrate the applicability of the 3DVAR algorithm to retrieve large-scale wind fields and meteorological wind maps in an area spanning from the Arctic and south to mid-Norway. Therefore, we expanded the domain of a geographic grid from 61° N to 81° N and from 6° E to 32° E. We implemented a 2 degree longitudinal increment and a 0.5° increment in latitude. Furthermore, we increased the coupling strength to the next neighbors by a factor of 4. The other settings were similar to those of the retrievals shown above.

This demonstration retrieval was performed with data from December 2012. During this time a unique configuration of MRs was operational in the Nordic countries and on Svalbard. In addition to the Nordic Cluster described previously, the Alta MR was in operation on Bear Island (Nozawa et al., 2012), which is located between the Norwegian mainland and Svalbard. The Bear Island radar was located close to the meteorological station on the island (74.475423° N, 19.207040° E). In addition, this was the first winter season that the Norwegian University of Science and Technology (NTNU) MR at Trondheim (63.406822° N, 10.467646°) was operated (de Wit et al., 2014; de Wit et al., 2015). We performed a similar quality control test for the Bear Island and Trondheim MRs as for all other radars and obtained vertical velocity residuals of about 2 cm/s and a variance of 15-22 cm/s (data not shown). During this period the Svalbard MR suffered from external interference and detected only 700-800 meteors per day, which resulted in a much lower measurement response.

Figure 16 shows two examples of zonal and meridional winds from the 3DVAR retrieval for December 2012. The wind fields are plotted at all grid cells independent of the width of the averaging kernels, which provides a better visualization of the large-scale flow and the associate high- and low-pressure systems indicated by the wind rotation. Although the wind fields are much smoother compared to the high-resolution retrievals, it is possible to see an increased variability and the presence of smaller scale dynamics above the radars, in particular, above the Nordic Meteor Radar Cluster consisting of TRO, KIR

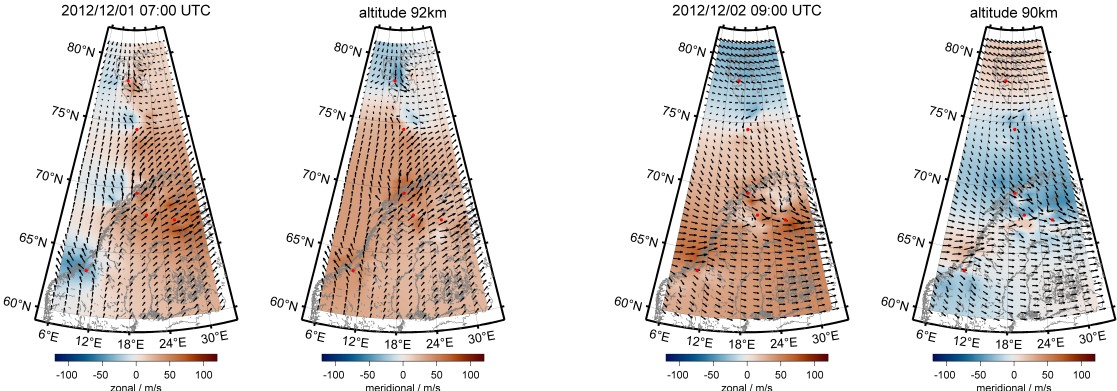

**Figure 16.** Meteorological wind maps obtained from the 3DVAR algorithm using a geographic grid and all MRs involved in ARISE.

and SOD at that time. The two snapshots also indicate that there are times with substantially different wind systems above the Arctic latitudes and the Norwegian mainland. There are remarkable differences in the horizontal winds between the two images, although only separated by two hours in time and 2 kilometers in altitude. This is mainly due to the semidiurnal tide, which is the dominating atmospheric wave during this time of the year at polar latitudes. Climatologies for December show amplitudes between 40-70 m/s and vertical wavelengths of about 30-40 km for the semidiurnal tide (Wilhelm et al., 2019; Stober et al., 2019). A semidiurnal tide leads to a clockwise rotation in the flow field, which can is found comparing the wind vectors between the two time steps.

When examining the measurement response for such retrievals, shown in Figure 17, it is straightforward to identify potential locations to improve the network coverage. One objective could be to connect the Trondheim MR to the other Nordic MRs on the mainland. The performed large scale retrieval made use of meteor radars that contributed to the Atmospheric Dynamics Research InfraStructure in Europe (ARISE) (Blanc et al., 2018). The Bear Island observations support a potential possibility to link the mainland systems up to the Svalbard region by using passive receivers similar to those used for CONDOR.

## 6 Discussion

Optimal estimation is an established mathematical technique to retrieve atmospheric quantities and results in statistically sound solutions to inverse problems based on a priori knowledge. During the past years, several multistatic meteor radar networks have been deployed. However, these observations are often analyzed using least squares techniques to infer horizontal inhomo-geneities in the wind fields (Stober and Chau, 2015; Chau et al., 2017), which are based on assumptions that are not necessarily valid. Wind speeds in the stratosphere and mesosphere are essentially fast enough to reach the subsonic compressible flow regime (Mach>0.3) and, thus, terms such as horizontal divergence have to be handled with care.

More recently, initial analysis applying inverse theory were presented by Stober et al. (2018a) and Chau et al. (2021). These

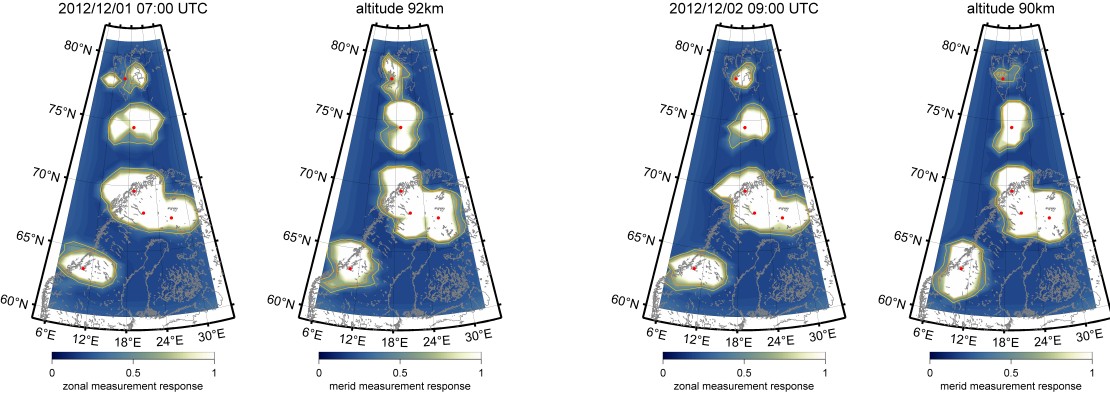

**Figure 17.** Measurement responses for the large-scale retrieval visualized with next-neighbor correlation corresponding to about 120 km resolved scales.

retrievals used a Tikhonov regularization to cure the ill-posed mathematical problem by assuming a certain smoothness or curvature and solve for wind vector solutions at pre-defined grid locations or grid cells. Chau et al. (2021) presented some results from a Peruvian meteor radar network using a mean curvature as a constraint for the solution, which in principle is almost identical to the VVP method (Waldteufel and Corbin, 1979), whereas Stober et al. (2018a) only assumed local correlation and retrieved arbitrary wind fields. However, both algorithms just solve the winds within a 2D-layer. The 3DVAR retrieval outlined herein, is the first algorithm based on Bayesian statistics and solves the tomographic problem as a full 3D solution including vertical coupling. The new algorithm is based on the Shannon information entropy (Shannon, 1948) and follows the approaches presented in Rodgers (2000) for the implementation of the optimal estimation.

Another benefit of the 3DVAR retrieval is the possibility to derive the measurement response and the averaging kernels. Both quantities provide essential information about the inverse problem and the reliability of the retrieved quantities and how much information is mixed in from either the a priori distribution or due to remote correlations. We presented some examples of measurement responses for CONDOR and the Nordic Meteor Radar Cluster that apparently indicate that not all wind components can be retrieved within the domains assuming just a next-neighbor correlation with equal reliability reflected by a reduced measurement response. The measurement response for forward scatter systems or multistatic passive links is much less homogeneous compared to monostatic systems. This is understandable in the context of the forward scatter ellipse, which basically results in a large area with a small or almost negligible measurement response for the horizontal winds in the region between the Tx and Rx, whereas the horizontal winds are difficult to retrieve only directly above a monostatic radar. The measurement response for CONDOR contains two separated patches of increased measurement response for the zonal and meridional components, which reflects the alignment of the passive receivers in the North and South direction.

The ASGARD software permits us to use different retrieval grids for the 3DVAR, which can be arbitrarily defined and use either a Cartesian spacing following the Earth's surface or a geographic grid that is regular in longitude and latitude. This

flexibility simplifies potential comparisons to other observations or model outputs for validation. Furthermore, large domains are possible that cover large parts of the globe. Also other data sets can be easily included, even if there is no direct overlap between the observation volumes.

Secondary waves and their resulting forcing on the MLT is an emerging topic (Vadas and Becker, 2018; Vadas et al., 2018).
The generation of non-primary waves is driven by large-scale body forces (Vadas and Fritts, 2001). The 3DVAR algorithm is capable of resolving such body forces and the resulting acceleration/deceleration of the mean flow as well as resolving the two counter-rotating vortices. The presented example corresponds to the scales described by Vadas and Fritts (2001) providing confidence in the analysis.

Furthermore, we derived horizontal wavelength spectra similar to those presented in Nastrom and Gage (1985). Our preliminary
analysis suggests that the transition between the vortical $k^{-3}$ and divergent modes $k^{-5/3}$ occurs at smaller scales compared to those observed in the troposphere. We found a change of the slope at scales around 80-120 km, which is at the edge of the resolution, but still at a larger scale than the a priori correlation length.

Finally, we want to mention some additional quality controls that are included in the retrieval. We carefully evaluated all systems for potential range offsets that correspond to altitude deviations between the radars within a network. For the Nordic
Meteor Radar Cluster, we performed seasonal analysis and looked for systematic deviations of characteristic seasonal structures such as the summer wind reversal in the zonal wind. A similar attempt was pursued for CONDOR, but at a much shorter time scale of 14 days by comparing the vertical diurnal tidal pattern between all systems. The interferometry and alignment of the antennas were validated by estimating the vertical wind biases. Some MRs indicate some skewness in the vertical wind histograms pointing at some systematic differences. However, all systems observed a residual vertical wind bias in the order of
a few cm/s (<10 cm/s) and a standard deviation of 12-30 cm/s for the monostatic analysis. The 3DVAR results indicate a much broader distribution, but still a very small offset of the mean. Due to the lack of independent validations or other observations, we treat these vertical winds as residual bias. The residual vertical wind bias seems to be in agreement with trace gas observations (Straub et al., 2012) and radar data (Vincent et al., 2019; Gudadze et al., 2019). However, these values show a substantial disagreement to other MR and multistatic MR vertical wind data measurements (Egito et al., 2016; Chau et al., 2021; Conte
et al., 2021).

## 7 Conclusions

In this study, we present a new 3DVAR retrieval algorithm of the ASGARD software, which is designed to infer 3D tomographic winds from multistatic observations. The retrieval is applied to two meteor radar networks - CONDOR in Chile at the
585 Andes and the Nordic Meteor Radar Cluster. The 3DVAR retrieval is based on an optimal estimation approach and Bayesian statistics. Thus, the retrieved winds are statistically sound solutions considering the a priori knowledge.

The possibility to perform 3D tomographic retrievals at the MLT on different geographic or Cartesian grids provides more flexibility for the validation of the obtained winds or to combine and extend the domain areas. Furthermore, the results pre-

sented indicate that the algorithm is fast enough to run continuously and still to permit the analysis of small scale structures.
The Keogram analysis and the ATW plots are suitable to investigate the large-scale dynamics and to provide a quick overview of the network performance.

The 3DVAR retrievals are capable of resolving counter-rotating vortices that might be related to body forces, which is demonstrated in a case study of a westward propagating GW above the CONDOR network. The algorithm resolved the two counter-rotating vortices and the acceleration direction of the body force. The spatial scales (horizontal and vertical) for the body force event correspond to the expected theoretical results from previous studies. We were also able to observe large-scale vortices of about 300 km in diameter, which would remain otherwise undiscovered.

Due to the optimal estimation implementation of the 3DVAR, it is possible to compute the measurement response for each time and altitude bin. This information is required to identify potential holes or gaps within existing networks that can be closed by installing additional passive systems or monostatic meteor radars. The measurement response of monostatic radars is optimal to ensure unbiased wind inversion and the most isotropic measurement response. Passive multi-static links are ideally used in combination with monostatic radars. Finally, it is worth considering that passive multi-static links alone show an unisotropic measurement response caused by the shape of the forward scatter ellipse that could lead to biases.

*Data availability.* The data is available upon request. Please contact Alexander Kozlovsky (alexander.kozlovsky@oulu.fi) for the Nordic Meteor Radar Cluster and Alan Liu (LIUZ2@erau.edu) for CONDOR to obtain the 3DVAR retrievals.

## Appendix A: Coordinate transformations

The appendix includes all geometric coordinate transformations that are used in the manuscript and are also presented in (Stober et al., 2018a). However, we corrected some typos present in the geodetic to ECEF coordinate transformation.

### A1  Geodetic to ECEF

Radars often are connected to a GPS receiver, which provides the geodetic coordinate in longitude ($\lambda$), latitude ($\phi$) and height ($z$). The height refers to the altitude of the radar above the WGS84 reference ellipsoid (National Imagery and Mapping Agency, 2000; Hofmann-Wellenhof et al., 1994). The WGS84 defines the semi-major axis of the Earth to be $a = 6378137.0$ m and a reciprocal of flattening of $f = 1/298.257223563$. The semi-minor axis is given by $b = 6356752.3142$ m. These quantities permit to derive the first eccentricity squared $e^2$, the second eccentricity squared $e'^2$ and the radius of Earth's curvature $N$ by

$$
\begin{aligned}
e^2 &= 2 \cdot f - f^2 \ , \\
e'^2 &= (a^2 - b^2)/b^2 \ , \\
N &= a/\sqrt{(1 - e^2 \sin(\phi)^2)} \ .
\end{aligned}
\tag{A1}
$$

Considering the WGS84 reference ellipsoid of the Earth, any given geodetic location defined by a longitude, latitude and height (above the WGS84 surface) can be transformed to the ECEF coordinates ($X_R, Y_R, Z_R$) by

$$
\begin{aligned}
X_R &= (N + z) \cdot \cos(\phi) \cos(\lambda) \ , \\
Y_R &= (N + z) \cdot \cos(\phi) \sin(\lambda) \ , \\
Z_R &= (N + z - e^2 N) \sin(\phi) \ .
\end{aligned}
\tag{A2}
$$

### A2  ECEF to Geodetic

The reverse transformation from ECEF to a geodetic longitude ($\lambda$), latitude ($\phi$) and height ($z$) is more challenging. Zhu (1993) presents a summary and short validation of several algorithms. The transformation below is based on Heikkinen (1982). The advantage of this algorithm is the small error and that it is valid from the Earth center (z=-6300 km) up to the height of geostationary orbits.

$$e'^2 = (a^2 - b^2)/b^2 \tag{A3}$$

$$F = 54b^2 z^2$$

$$G = r^2 + (1 - e^2)z^2 - e^2(a^2 - b^2)$$

$$c = e^4 F r^2 / G^3$$

$$s = \sqrt[3]{1 + c + \sqrt{c^2 + 2c}}$$

$$P = \frac{F}{3(s + 1/s + 1)^2 G^2}$$

$$Q = \sqrt{1 + 2e^4 P}$$

$$r_0 = -\frac{Pe^2 r}{1 + Q} + \sqrt{\frac{a^2}{2}\left(1 + \frac{1}{Q}\right) - \frac{P(1 - e^2)z^2}{Q(1 + Q)} - \frac{Pr^2}{2}}$$

$$U = \sqrt{(r - e^2 r_0)^2 + z^2}$$

$$V = \sqrt{(r - e^2 r_0)^2 + (1 - e^2)z^2}$$

$$z_0 = \frac{b^2 z}{aV}$$

$$h = U\left(1 - \frac{b^2}{aV}\right)$$

$$\phi = \arctan((z + e'^2 z_0)/r)$$

$$\lambda = 2\arctan\left(\frac{\sqrt{X_R^2 + Y_R^2} - X_R}{Y_R}\right)$$

## A3  ENU to ECEF

An object observed at a distance $R$ from a known geodetic location in ENU coordinates at a certain azimuth and off-zenith angle can be converted to its ECEF coordinates using the following transformation. Here, the ENU coordinates $(x_m, y_m, z_m)$ are given at a geodetic longitude ($\lambda$), latitude ($\phi$) and height ($z$). Rotating the ENU vector $(x_m, y_m, z_m)$ into the ECEF reference $(X_m, Y_m, Z_m)$ and adding the radar vector results in;

$$\begin{bmatrix} X_m \\ Y_m \\ Z_m \end{bmatrix} = \begin{bmatrix} -\sin(\lambda) & -\sin(\phi)\cos(\lambda) & \cos(\phi)\cos(\lambda) \\ \cos(\lambda) & -\sin(\phi)\sin(\lambda) & \cos(\phi)\sin(\lambda) \\ 0 & \cos(\phi) & \sin(\phi) \end{bmatrix} \cdot \begin{bmatrix} x_m \\ y_m \\ z_m \end{bmatrix} + \begin{bmatrix} X_r \\ Y_r \\ Z_r \end{bmatrix}. \tag{A4}$$

## A4  ECEF to ENU

The conversion of ECEF to ENU coordinates is given by the line of sight vector from the radar towards the meteor in the frame of the ENU coordinates at the geodetic location of the meteor. If the line of sight velocity vector is observed at a certain azimuth $az$ and zenith $ze$ angle relative to the radar, it has a different azimuth $az'$ and zenith $ze'$ angle in the frame of the local geodetic coordinates of the meteor. Assuming that we know the ECEF coordinates of the meteor $(X_m, Y_m, Z_m)$ and the radar

location $(X_R, Y_R, Z_R)$ it is straightforward to compute the ENU coordinates $(x, y, z)$ by using;

$$
\quad \begin{bmatrix} x \\ y \\ z \end{bmatrix} = \begin{bmatrix} -\sin(\lambda) & \cos(\lambda) & 0 \\ -\sin(\phi)\cos(\lambda) & -\sin(\phi)\sin(\lambda) & \cos(\phi) \\ \cos(\phi)\cos(\lambda) & \cos(\phi)\sin(\lambda) & \sin(\phi) \end{bmatrix} \cdot \begin{bmatrix} X_m - X_R \\ Y_m - Y_R \\ Z_m - Z_R \end{bmatrix} . \tag{A5}
$$

The local azimuth $az'$ and $ze'$ with respect to the geodetic position of the meteor are obtained by;

$$
\begin{aligned}
az' &= \arctan(y/x) \\
ze' &= \arccos\left(\frac{z}{\sqrt{x^2+y^2+z^2}}\right) .
\end{aligned} \tag{A6}
$$

*Author contributions.* GS developed ASGARD and the 3DVAR algorithm. AL, AK and QZ supported the development and implementation
of the software at CONDOR and the Nordic Meteor Radar Cluster. JK, EB, CH, MT, SN, PE, RH, NM and ML provided meteor radar data from the networks and supported the implementation of the software. All authors contributed to the editing of the manuscript.

*Competing interests.* Gunter Stober, Jorge L. Chau and Juha Vierinen submitted a patent for multi-static meteor radar networks using pulsed and cw-radars.

*Acknowledgements.* Gunter Stober is a member of the Oeschger Center for Climate Change Research (OCCR) at the University Bern. Most
665 of this software was developed during a sabbatical at the University of Western Ontario. Gunter Stober is grateful to Prof. Peter Brown and Prof. Robert Sica for the support and discussions during this visit. for The authors also want to thank Chris Adami and Ian Reid from ATRAD Ltd, and Brian Fuller and Adrian Murphy from Genesis Ltd for valuable discussions about some technical aspects of the deployed radars and software. Mark Lester is supported by STFC grant ST/S000429/1. The Esrange meteor radar operation, maintenance and data collection is provided by Esrange Space Center of Swedish Space Corporation. The 3DVAR retrievals were developed as part of
670 the ARISE design study (http://arise-project.eu, last access: October 2020) funded by the European Union's Seventh Framework Programme for Research and Technological Development. Research Council of Norway under the project Svalbard Integrated Arctic Earth Observing System—Infrastructure development of the Norwegian node (SIOS-InfraNor, Project No. 269927). This study is partly supported by Grants-in-Aid for Scientific Research (17H02968) of Japan Society for the Promotion of Science (JSPS). GS thanks Andreas Dörnbrack from DLR Oberpfaffenhofen for valuable discussions and helpful comments.

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
