# Peer review of "Atmospheric tomography using the Nordic Meteor Radar Cluster and Chilean Observation Network De Meteor Radars: network details and 3DVAR retrieval"

_Atmospheric Measurement Techniques, 2021_

## Author Comment (AC1)

histogram of residual bias vertical velocities
Esrange (KIR)

residual vertical velocity bias
mean=0.044 m/s
std=0.166 m/s

Esrange (KIR)

[Figure]

histogram of residual bias vertical velocities
ICON-UA Esrange (KIR)

residual vertical velocity bias
mean=0.003 m/s
std=0.152 m/s

ICON-UA Esrange (KIR)

[Figure]

histogram of residual bias vertical velocities
Sodankyla (SOD)

residual vertical velocity bias
mean=0.136 m/s
std=0.228 m/s

Sodankyla (SOD)

[Figure]

histogram of residual bias vertical velocities
ICON-UA Sodankyla (SOD)

residual vertical velocity bias
mean=0.002 m/s
std=0.153 m/s

ICON-UA Sodankyla (SOD)

---

## Author Response (AR1)

Point-by-point reply to reviewer #1:

**General Comment:**

Review of "Atmospheric tomography using the Nordic Meteor Radar Cluster and Chilean Observation Network De Meteor Radars: network details and 3DVAR retrieval" by Stober et al.

The paper introduces a new technique to derive 3D wind fields from networks of meteor radar stations in a tomographic approach. The performance of this 3DVAR retrieval is demonstrated for two meteor radar networks, the Nordic Meteor Radar Cluster in Northern Europe, and the CONDOR network in South America.

Based on several observed events, the benefits of this approach, and the different characteristics of the two radar networks are discussed. As a diagnostic parameter, the Shannon information content is derived. It is found that, as a consequence of its linear arrangement, the CONDOR network is more sensitive to meridional winds over the line connecting the stations, and more sensitive to zonal winds in two patches of enhanced sensitivity parallel to this line.

It is shown that both radar networks are capable to resolve the counter-rotating vortices of breaking gravity wave events that are important for the excitation of secondary waves, which is currently a hot topic in atmospheric dynamics. Horizontal wavelength spectra are derived, and the impact of a minor sudden stratospheric warming in December 2019 is investigated in a keogram analysis.

Overall, the paper is very well written and fits in the scope of AMT.

Publication of the paper in AMT is therefore recommended after addressing my minor comments.

**General Reply:**

We thank the reviewer for his valuable comments and suggestions. We considered the main comments in the revision. All changes are tracked by latexdiff and are indicated by color codes. We expanded the discussion of the GW analysis and added additional panels to support our findings. However, the main emphasis of this paper is the retrieval algorithm rather than a detailed GW analysis. We are working on further publications looking in more detail on the data.

MAIN COMMENTS:

**Comments:**

(1) The difference between the absolute wind speeds in Fig.4, left, and Fig.4, right, should be calculated and discussed. This difference can be used as further diagnostics and measure of errors.

(2) The Data Availability section of the paper is missing.

**Reply:**

Both main comments are answered in detail below.

SPECIFIC COMMENTS:

**Comment :**

(1) l.29: The reference de Wit et al. (2017) should be included here as an earlier reference for the occurrence of secondary gravity waves over South America.

de Wit, R. J., D. Janches, D. C. Fritts, R. G. Stockwell, and L. Coy (2017),

Unexpected climatological behavior of MLT gravity wave momentum flux in the lee of the Southern Andes hot spot,

Geophys. Res. Lett., 44, 1182-1191, doi:10.1002/2016GL072311.

(2) l.37: There are also imaging satellite instruments that provide spatially resolved 2D, or even 3D observations of gravity waves. These should be mentioned here:

Randall, C. E., et al. (2017),

New AIM/CIPS global observations of gravity waves near 50-55 km,

Geophys. Res. Lett., 44, 7044-7052, doi:10.1002/2017GL073943.

Ern, M., L. Hoffmann, and P. Preusse (2017),

Directional gravity wave momentum fluxes in the stratosphere derived from high-resolution AIRS temperature data,

Geophys. Res. Lett., 44, 475-485, doi:10.1002/2016GL072007.

**Reply:**

We expanded the introduction as suggested and added the references as proposed.

**Comment:**
(3) Caption of Fig.4 is incomplete. Suggestion:

...Cluster and a Cartesian geographic grid.

->

...Cluster using a Cartesian grid (left) and a longitude/latitude geographic grid (right).

**Reply:**

Done.

**Comment:**
(4) Fig.4:

Please show in an additional panel the wind strength difference between the two representations, and add some discussion. This would give the reader an impression of the robustness of the results. At least in the regions of high measurement content the differences should be small.

**Reply:**

We thank the reviewer for this comment and added the requested panel to the Figure. We also updated the central panel of the geographic retrieval with the latest version of the software, where certain issues at the domain boundary were ironed out. We had to compute a new geographic picture to match the voxel size and correlation length without introducing artefacts. We also added a short paragraph describing the additional results.

**Comment:**

(5) Fig.5: Please comment!

Is this a typical event, or a particularly strong event?

**Reply:**

The image represents a typical retrieval result. The wind magnitudes in this image are driven by a diurnal tide. We added this information in the text.

**Comment:**
(6) l.354: Please add this information:

Do you assume vertical wind to be zero for the data assimilation mode?

**Reply:**

Yes, we added this information explicitly in the paragraph.

**Comment:**

(7) Caption of Fig.8 does not match the figure!

Shown is the measurement response, not the wind fields.

**Reply:**

Corrected.

**Comment:**

(8) l.408/409: Here you state that gravity wave activity would be enhanced at 69-70N. Please be more specific!

Does this statement refer to Fig.10 where stronger variability is seen in the two panels on the right hand side?

However, the left two panels are domain-averages and should therefore be much smoother, anyhow.

Or does this refer to Fig.11?

In Fig.11 the semidiurnal tide is the strongest mode of variability, and other fluctuations are difficult to see. Could you therefore provide some more guidance to the reader where exactly one should see this effect?

**Reply:**

We added additional panels to the figures to highlight potential GW activity and differences between the domain mean and the local observations. Due to another suggestion by reviewer 2, we also added the decomposition of the time series into daily mean winds and semidiurnal tidal amplitudes. Furthermore, we expanded the discussion and description of the additional panels in the paragraph and throughout the section. However, a detailed analysis of GW is beyond the scope of this paper and might would destroy the readability in some parts.

**Comment:**

(9) Fig.14: Please add information!

The wind fields are very different. Please state whether this is an effect of the semidiurnal tide.

**Reply:**

We added the following text:

**There are remarkable differences in the horizontal winds between the two images, although only separated by two hours in time and 2 kilometers in altitude. This is mainly due to the semidiurnal tide, which is the dominating atmospheric wave during this time of the year at polar latitudes. Climatologies for December show amplitudes between 40-70 m/s and vertical wavelengths of about 30-40 km for the semidiurnal tide \citep{Wilhelm:2019,Stober_2019-1006_NAVGEM}. A semidiurnal tide leads to a clockwise rotation in the flow field, which can is found comparing the wind vectors between the two time steps.**

**Comment:**
(10) Fig.14: Question: are the "92km" for the upper left two panels wrong?

Should it read "90km"?

**Reply:**

Thank you for spotting this mistake. The revised manuscript now only contains zonal and meridional winds and measurement response for the 90 km level. There is only very little difference in the measurement response between 90 and 92 km altitude. But, we want to demonstrate that we can run these large scale retrievals also at other altitude then just 90 km.

**Comment:**
(11) The "Data availability" section is missing!

**Reply:**

We added the following statement: 'The data is available upon request. Please contact Alexander Kozlovsky (alexander.kozlovsky@oulu.fi) for the Nordic Meteor Radar Cluster and Alan Liu (LIUZ2@erau.edu) for CONDOR to obtain the 3DVAR retrievals.'

However, the instrument PI's, which all of them are co-authors, own the data and should be involved in any further data exchange. The 3DVAR retrievals are supposed to be uploaded to the ARISE-IA data base, but due to the lack of funding the project got delayed.

TECHNICAL COMMENTS:

**Comment:**

caption of Table 1: "ALO" does not belong to the Nordic meteor radars.

Nordic meteor radars -> Nordic and ALO meteor radars

**Reply.**

Done.

**Comment:**
l.251: to included -> to include

**Reply:**

Done.

**Comment:**
l.418: periodigrams -> periodograms  ???

(periodigrams is rarely used)

**Reply:**

Done.

**Comment:**
l.486: Peruian -> Peruvian

**Reply:**

Done.

Point-by-point reply to reviewer #2:

**General Comment:**

The paper introduces a new algorithm to analyze the MLT observations by the radar networks and utilizes two the data from two radar networks to demonstrate the results based on this data retrieval algorithm. The new and improved capability to obtain high quality horizontal mapping of zonal and meridional winds by this technique is quite impressive. The paper has demonstrated that this work is high quality and provide a new tool to study various important dynamic topics in the mesosphere and lower thermosphere in different scales. The figures are clean and clear with proper captions. I understand this is more like a technology journal, but it would be good for illustrate how some of the dynamic parameters was calculated. For example, the body force. The section of discussion reads like a summary of this work.

**General reply:**

We thank the reviewer for his valuable and constructive comments on our manuscript. The paper is revised according to the suggestions and we prepared a detailed response to all the raised points. However, we still did not perform a detailed geophysical analysis of the various events as this is beyond the scope of this paper. The manuscript is supposed to provide a detailed outline of the algorithm and demonstration of the capabilities to observe certain meteorological phenomena such as body forces, large-scale vortices or large-scale retrievals. The geophysical analysis of all these possibilities is not possible within the frame of this publication and would make the paper much more difficult to read. Therefore, we added as suggested some more geophysical content, but leave the detailed analysis to later publications.

Reply to minor and technical concerns:

**Comment:**

Line 38. The Na Doppler lidar has been an important ground-based instrument, and has many important contributions to the MLT dynamics, due to its capability of day and night time simultaneous measurements of temperature and horizontal winds [Krueger et al., 2015]. Due to its horizontal wind capability, it can derive the intrinsic properties of GWs. It is unfortunate that the author misses this important instrument.

**Reply:**

We added a few sentences in the introduction explicitly referring to Na-lidars including the proposed citation and also mention different analysis methods to extract the intrinsic gw parameters. Resonance lidars have indeed played an important role in understanding physical processes at the MLT. There was no intention to exclude or not mentioning a certain lidar type.

**Comment:**

Line 44-45. There are actually many of this collaborative investigations between the Na Doppler lidar and airglow instrument. So, I suggest the author replace "only a few" with many. The problem with such Na Doppler lidar – Airglow investigations is that they are

mostly focusing on single case studies, such as Yuan et al. 2016, Cai et al., 2014, and, thus, cannot provide statically large numbers of cases to build robust database of these intrinsic properties of GWs.

**Reply:**

We rephrased the sentence to avoid the ambiguity in the understanding of a few. We refer the 'a few' statement to the number of observatories or research facilities. The new sentence reads: "There are only a few observatories or research facilities in the world with a unique suite of simultaneous common volume observations of winds, temperatures and airglow to determine intrinsic GW properties. However, these observations led to many collaborative studies (many citations)"

**Comment:**

Line 2. Replace "can be" with "is"

**Reply:**

Done.

**Comment:**

Line 60, to investigate

**Reply:**

Done.

**Comment:**

Line 73, delete "are going to"

**Reply:**

Done.

**Comment:**

Line 224, demands

**Reply:**

Done.

**Comment:**

Line 231, delete "thus"

**Reply:**

Done.

**Comment:**

Line 382, replace "possibilities" with "capabilities"

**Reply:**

Done.

**Comment:**

Line 384, delete "very"

**Reply:**

Done.

**Comment:**

Line 401-409, the semidiurnal tidal activity is not quite clear in Fig. 10. I wonder if the author can derive and show the semidiurnal amplitude variations instead of the hourly winds.

**Reply:**

We are added two more panels showing daily mean winds and daily mean semidiurnal amplitudes. The discussion of the Figure is expanded concerning these results.

**Comment:**

Line 440, I am not quite clear what depend on the choice of… Please specify.

**Reply:**

We rephrased this part. The Lagrange multiplier in the current version is set to 1 and, thus, only the apriori covariance plays a role. However, the strength or weight of the cost function term can be modified by increasing/decreasing the Lagrange multiplier.

**Comment:**

Line 440 – 443. This is a very long sentence, please consider to revise.

**Reply:**

We changed the wording and separated the sentence into several shorter.

**Comment:**

Line 521, I do not see any "diurnal tidal pattern" is discussed in the paper.

**Reply:**

We added a figure for the ATW for CONDOR showing a pronounced diurnal tidal pattern.

---

## Author Response (AR2)

General reply:

We thank both reviewers and the editor for the positive evaluation of the manuscript.